# Uterine adenomyosis is an oligoclonal disorder associated with *KRAS* mutations

Satoshi Inoue [1,2]*, Yasushi Hirota [3], Toshihide Ueno [2], Yamato Fukui[3], Emiko Yoshida[4], Takuo Hayashi[5], Shinya Kojima[2], Reina Takeyama[2], Taiki Hashimoto [6], Tohru Kiyono [7], Masako Ikemura[8], Ayumi Taguchi [3], Tomoki Tanaka[3], Yosuke Tanaka[2], Seiji Sakata[9], Kengo Takeuchi [9,10,11], Ayako Muraoka[12], Satoko Osuka[12], Tsuyoshi Saito[5], Katsutoshi Oda[3], Yutaka Osuga[3], Yasuhisa Terao [4], Masahito Kawazu[1,2] & Hiroyuki Mano[2]

Uterine adenomyosis is a benign disorder that often co-occurs with endometriosis and/or leiomyoma, and impairs quality of life. The genomic features of adenomyosis are unknown. Here we apply next-generation sequencing to adenomyosis (70 individuals and 192 multi-regional samples), as well as co-occurring leiomyoma and endometriosis, and find recurring *KRAS* mutations in 26/70 (37.1%) of adenomyosis cases. Multi-regional sequencing reveals oligoclonality in adenomyosis, with some mutations also detected in normal endometrium and/or co-occurring endometriosis. *KRAS* mutations are more frequent in cases of adeno-myosis with co-occurring endometriosis, low progesterone receptor (PR) expression, or progestin (dienogest; DNG) pretreatment. DNG's anti-proliferative effect is diminished via epigenetic silencing of *PR* in immortalized cells with mutant *KRAS*. Our genomic analyses suggest that adenomyotic lesions frequently contain *KRAS* mutations that may reduce DNG efficacy, and that adenomyosis and endometriosis may share molecular etiology, explaining their co-occurrence. These findings could lead to genetically guided therapy and/or relapse risk assessment after uterine-sparing surgery.

[1] Department of Medical Genomics, The University of Tokyo, Tokyo 113-0033, Japan. [2] Division of Cellular Signaling, National Cancer Center Research Institute, Tokyo 104-0045, Japan. [3] Department of Obstetrics and Gynecology, The University of Tokyo, Tokyo 113-0033, Japan. [4] Department of Obstetrics and Gynecology, Juntendo University Faculty of Medicine, Tokyo 113-8421, Japan. [5] Department of Human Pathology, Juntendo University Faculty of Medicine, Tokyo 113-8421, Japan. [6] Division of Diagnostic Pathology, National Cancer Center Hospital, Tokyo 104-0045, Japan. [7] Division of Carcinogenesis and Cancer Prevention, and Department of Cell Culture Technology, National Cancer Center Research Institute, Tokyo 104-0045, Japan. [8] Department of Pathology, Graduate School of Medicine, The University of Tokyo, Tokyo 113-0033, Japan. [9] Pathology Project for Molecular Targets, The Cancer Institute, Japanese Foundation for Cancer Research, Tokyo 135-8550, Japan. [10] Clinical Pathology Center, The Cancer Institute, Japanese Foundation for Cancer Research, Tokyo 135-8550, Japan. [11] Division of Pathology, The Cancer Institute, Japanese Foundation for Cancer Research, Tokyo 135-8550, Japan. [12] Department of Obstetrics and Gynecology, Nagoya University Graduate School of Medicine, Nagoya 466-8550, Japan. *email: satoshiinouencc4@gmail.com

Uterine adenomyosis is a benign condition in which endometrium-like epithelial and stromal tissues appear in the myometrium, surrounded by hypertrophic smooth muscle. Similar to endometriosis and leiomyoma, adenomyosis is common in women of reproductive age[1,2]. Adenomyosis and endometriosis are closely linked diseases, as both are estrogen-dependent and often co-exist in the same patients[3–5]. This co-occurrence is a risk factor for relapse in women who have undergone fertility sparing surgery[6–8].

Although adenomyosis and endometriosis share some histological features and molecular changes[9], there are notable differences in their pathogenesis, lesion localization, and clinical features[10,11]. For example, menorrhagia, recurrent implantation failure, and pregnancy loss are frequent in adenomyosis but not in endometriosis[12,13]. In addition, multiparity and prior uterine surgeries are linked to adenomyosis but not to endometriosis[14]. Thus, adenomyosis has come to be recognized as a distinct clinical entity.

The etiology of adenomyosis is currently unknown. Several theories for its development have been proposed, including direct invasion of endometriotic cells or an endometriotic lesion, metaplasia of displaced embryonic pluripotent Mullerian remnants, and differentiation of endometrial stem/progenitor cells[5,15]. In addition, despite the known differences between endometriosis and adenomyosis, Koninckx et al.[9] hypothesized in a recent review that a common molecular mechanism might underlie the pathogenesis of these disorders. However, no direct evidence to support this hypothesis exists at present, due to a lack of knowledge on the molecular mechanisms underlying adenomyosis pathogenesis.

The growth of an adenomyotic lesion is generally enhanced by estrogen and abrogated by progestogens. Adenomyotic symptoms are currently relieved using either endocrine therapies based on gonadotrophin-releasing hormone agonist (GnRHa) or progestins such as dienogest (DNG), or by employing a levonorgestrel-releasing intrauterine system (LNG-IUS)[16,17]. However, no formal guidelines for best medical management using these hormonal therapies exist[16]. Identification of the molecular underpinnings of adenomyosis could offer opportunities for precision treatment of this condition.

Next-generation sequencing (NGS) is a powerful tool that permits the identification of somatic genomic alterations. Whereas mutations in *MED12* are frequent in leiomyoma (~70%)[18], recurrent mutations in cancer-associated genes such as *KRAS*, *PIK3CA*, *ARID1A*, *PPP2R1A*, *PTEN*, and/or *TP53* occur in uterine endometrial carcinoma[19]. Similarly, anatomical subtypes of endometriosis, such as ovarian endometrioma (EN-OV) and deep infiltrating (EN-DI) endometriosis, harbor mutations in *PIK3CA*, *ARID1A*, *PPP2R1A*, and/or *KRAS*[20,21]. Importantly, *PIK3CA*- or *KRAS*-mutated clones arising in histologically normal uterine endometrium have been proposed as the cellular origin of endometriosis[21]. However, very recent genomic analyses of normal human tissues, including endometrium, have undermined this hypothesis. RNA-sequencing analyses of 29 types of normal tissues, including the uterus, have uncovered the presence of expanded clones of cells with relatively high mutation burdens in many normal tissues[22]. Furthermore, endometrium samples from individuals who have undergone endometrial biopsy for a reason unrelated to endometriosis have frequently shown *KRAS* and *PIK3CA* mutations[23,24]. Thus, endometrial clones bearing *KRAS* and/or *PIK3CA* mutations may not necessarily drive the pathogenesis of endometriosis. Ideally, the status of *KRAS*- and/or *PIK3CA*-mutated clones in histologically normal endometrium (NE) should be compared between individuals with and without endometriosis. Accordingly, the exact contribution of *KRAS* and *PIK3CA* mutations to endometriosis remains unresolved.

In contrast to endometriosis, there has been, to date, no parallel genomic analysis of adenomyosis. Therefore, it is unknown whether adenomyosis involves clonal proliferation and whether its mode of molecular pathogenesis is shared with other gynecological disorders. This highlights the need for a comprehensive genomic characterization of adenomyosis to provide insights into many important and unresolved questions in adenomyosis etiology and pathology. To address this knowledge gap, we conduct NGS on a large panel of adenomyosis and co-occurring leiomyoma and endometriosis tissues. Our analyses reveal the presence of oligoclonality and recurrent *KRAS* mutations in adenomyosis tissues, and suggest that adenomyosis and endometriosis share molecular etiology, which may explain their frequent co-occurrence. Furthermore, we provide functional evidence for the role of mutant *KRAS* in mediating the efficacy of DNG as an adenomyosis therapy. Importantly, our findings could inform relapse risk assessment and therapeutic strategies for adenomyosis patients.

## Results

**Somatic mutations are present in adenomyosis.** To define the molecular pathology of adenomyosis, we used NGS to characterize its genomic landscape. We collected fresh-frozen samples from 70 individuals: a discovery cohort of 51 adenomyosis patients whose samples were subjected to whole exome sequencing (WES), plus an additional 19 individuals who were biopsied at a later time and whose samples were subjected to targeted deep sequencing (TDS) (Supplementary Data 1 and 2). In 29 of these 70 individuals, multi-regional sampling of adenomyosis with/without co-existing endometriosis and leiomyoma was performed and adjacent normal tissues were collected (Supplementary Fig. 1 and Supplementary Data 3). In total, we banked fresh-frozen lesions of adenomyosis (192 specimens from 70 individuals), endometriosis (15 specimens from 10 individuals), leiomyoma (13 specimens from 10 individuals), ovarian cancer (5 specimens from 5 individuals), adjacent normal myometrial tissues (13 specimens from 12 individuals), and adjacent normal endometrial tissues (8 specimens from 6 individuals). As germline controls, we collected fresh-frozen peripheral blood or mononuclear cells from ascites (70 specimens from 70 individuals) (Supplementary Data 3). WES was performed on these samples with coverage of 130–170× for adenomyosis, endometriosis, leiomyoma, and ovarian cancer, and ~100× for adjacent and normal tissue samples (Supplementary Fig. 2 and Supplementary Data 4). Our robust criteria and validation by TDS (see Methods, Supplementary Fig. 3, and Supplementary Data 5) permitted us to detect 134 unique synonymous and non-synonymous single-nucleotide variations (SNVs) in 31/51 (60.8%) adenomyosis cases (Supplementary Data 6), raising the possibility that adenomyosis is a clonal disorder with somatic mutations. These adenomyosis SNVs were present in low numbers (mean of 2.6 mutations/individual) and at low variant allele frequencies (VAF) (mean 4.8%; range 2.47–16.02%). These VAF values are comparable to those in co-occurring endometriosis but much lower than those in co-occurring leiomyoma and ovarian cancer (Fig. 1a, b, Supplementary Figs. 4 and 5, and Supplementary Data 6–9).

**KRAS is recurrently mutated in adenomyosis.** Analysis of the SNVs in adenomyosis cases revealed recurrent mutations encoding oncogenic KRAS p.G12 alterations (Patients #2, #3, and #6 in Supplementary Data 6). Patient #2 also harbored a mutation encoding the PPP2R1A p.R179 alteration observed in endometrial cancer and endometriosis. However, other mutations associated with uterine endometrial carcinoma and/or endometriosis (e.g., *PIK3CA*, *ARID1A*, *PTEN*, and *TP53*)[19–21] were not

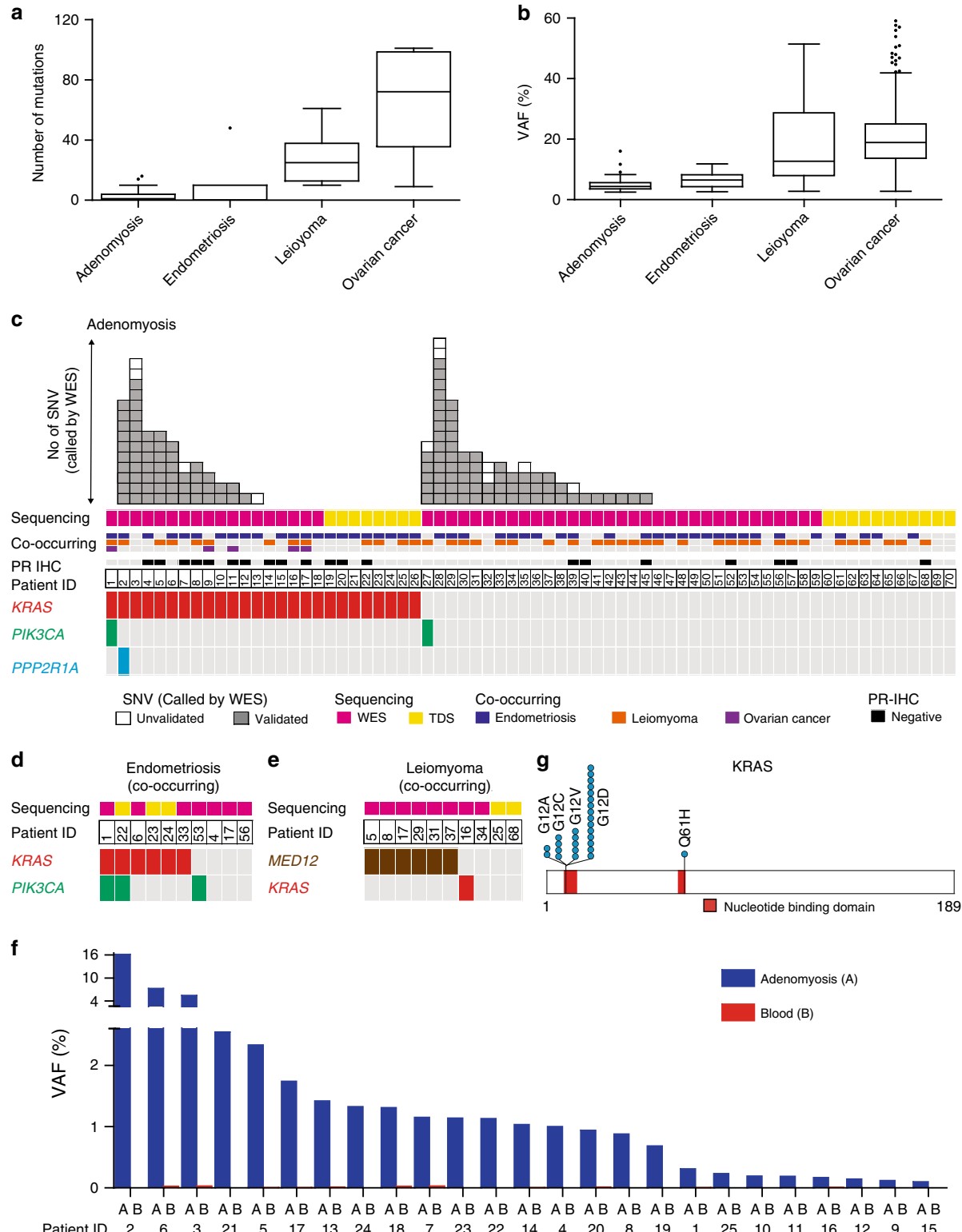

identified in adenomyosis using our SNV criteria (Supplementary Data 6). We speculated that, confined by our criteria (mutant read number <8) (Supplementary Fig. 3), our WES coverage may not have been deep enough to detect alterations in these genes that had low VAF due to either minimal content or limited expansion of mutated adenomyosis clones. To sharpen our analysis, we examined the ClinVar database (https://www.ncbi.nlm.nih.gov/clinvar/), a public database of pathogenic gene variations,

to determine whether any registered pathogenic/driver mutations of *KRAS*, *PIK3CA*, *PPP2R1A*, *ARID1A*, *PTEN*, and/or *TP53* were detectable in our adenomyosis WES data, including in the small number of mutant reads that had been filtered out by our SNV criteria. Using this strategy, we identified low numbers (<4) of pathogenic mutations of *KRAS*, *PIK3CA*, *PPP2R1A*, *ARID1A*, *TP53*, and *PTEN* in adenomyosis samples (Supplementary Data 10). To explore whether these candidate mutations were

**Fig. 1 Number and allele frequency of single-nucleotide variations (SNVs) identified in adenomyosis and co-occurring lesions. a** Number of somatic mutations in samples of adenomyosis ($n = 51$) and co-occurring cases of endometriosis ($n = 7$), leiomyoma ($n = 8$), and ovarian cancer ($n = 5$) presented as Tukey's Box-and-Whisker plots. **b** Variant allele frequency (VAF) identified in the adenomyosis and co-occurring endometriosis, leiomyoma, and ovarian cancer samples in **a** presented as Tukey's Box-and-Whisker plots. **c–e** Recurrence of the indicated gene mutations in adenomyosis (**c**), co-occurring endometriosis (**d**), and co-occurring leiomyoma (**e**). Top boxes: each box represents an individual SNV that was identified by whole exome sequencing (WES). SNVs were validated (grey) or not (white) by targeted deep sequencing (TDS). Middle boxes: samples were subjected to WES (purple) or TDS (yellow). Adenomyosis patients with co-occurring endometriosis (blue), leiomyoma (orange), and/or ovarian cancer (purple), and low progesterone receptor (PR) expression (black) are indicated above the patient number. Bottom boxes: each box represents an individual mutation of *KRAS* (red), *PIK3CA* (turquoise), *PPP2R1A* (blue), or *MED12* (brown). **f** VAF values for *KRAS* mutations in adenomyosis lesions (blue) or peripheral blood (red) as determined by TDS. **g** Schematic representation of somatic mutations in KRAS.

truly SNVs or pseudo-positives, we performed TDS on the adenomyotic lesions as well as on the corresponding germline control samples wherever possible. Although no mutation candidates in *ARID1A*, *TP53*, or *PTEN* were validated as somatic mutations in adenomyosis (Supplementary Data 11), this TDS analysis revealed recurrent somatic pathogenic mutations of *KRAS* and *PIK3CA* in this disease (Supplementary Data 12).

We next applied TDS analysis of *KRAS*, *PIK3CA*, and *PPP2R1A* to not only the discovery cohort ($n = 51$; Fig. 1c, pink sequencing column) but also to the additional 19 adenomyosis patients (Fig. 1c, yellow sequencing column). Recurrent somatic pathogenic *KRAS* mutations, including 25 cases of alterations at G12 and 1 case at Q61, were identified in 37.1% (26/70) of adenomyosis cases (Figs. 1c, f, g and Supplementary Data 12). Somatic mutations encoding the PIK3CA p.H1047 alteration were validated in lesions from Patients #1 and #27, as was a mutation encoding PPP2R1A p.P179 in a lesion from Patient #2 (Fig. 1c and Supplementary Data 12). Based on these findings, we hypothesized that somatic *KRAS* mutation might be an important genomic alteration associated with adenomyosis. Consistent with published data[18,20,21], samples of co-occurring endometriosis in some of our adenomyosis patients bore both *KRAS* and *PIK3CA* mutations (Fig. 1d and Supplementary Data 12), whereas most co-occurring leiomyoma samples harbored *MED12* mutations (Fig. 1e). Thus, the profiles of the dominant mutations linked to these three benign gynecological disorders differ.

**Mutations in the epithelial cell component of adenomyosis.** Only a minor fraction of an adenomyosis lesion comprised epithelial cells (Fig. 2a). This observation prompted us to consider whether the low VAFs we observed were due to low epithelial component content or limited clonal expansion of mutated cells. To isolate the epithelial component of adenomyosis samples, we performed laser capture microdissection (LCM) on formalin-fixed paraffin-embedded (FFPE) tissue samples from Patients #2, #6, and #8 (Fig. 2b–e). We isolated genomic DNA from epithelial cells of adenomyosis tissue (LCM-A), as well as from adjacent muscle cells (non-diseased control; LCM-ADJ), and performed TDS. Compared with bulk frozen samples, VAFs were markedly increased in LCM-A (but not LCM-ADJ) samples from all three adenomyosis cases (Fig. 2c–e, Supplementary Fig. 6, and Supplementary Data 13), demonstrating that the low VAFs detected in our WES analyses of bulk frozen adenomyosis lesions were due to low epithelial component content rather than to poor expansion of mutated adenomyosis clones. These LCM experiments indicate that somatic mutation occurs in the epithelial component of adenomyosis, and that adenomyosis may thus arise from the ectopic proliferation of mutated epithelial cell clones.

**Adenomyosis is clonally diverse and distinct from leiomyoma.** To initiate our investigation of clonal diversity within adenomyosis lesions, we performed multi-regional sampling and TDS of adenomyosis and co-existing leiomyoma lesions from Patients

#28, #29, and #5. We detected multiple mutations in each lesion (Fig. 3, Supplementary Figs. 7–9, and Supplementary Data 14–16). Some mutations were shared among almost all multi-regional samples from a single patient (e.g., *ZNF672* in Patient #28 (Fig. 3a); *C1QTNF* and *MSS51* in Patient #29 (Fig. 3b)), suggesting that these genetic alterations were acquired early during adenomyosis development, whereas other mutations were more restricted (Fig. 3). Most mutations detected in adenomyosis and co-existing leiomyoma were mutually exclusive (Figs. 3b, c and 4a, and Supplementary Data 15–17), implying the lack of a clonal relationship between these disorders. Thus, despite their frequent co-occurrence, adenomyosis and leiomyoma are distinct entities.

***KRAS*-mutated clones in adenomyosis occur in normal tissues.** To investigate whether the genomic alterations in adenomyosis originate in histologically NE and/or myometrium (NM), we compared mutations detected by TDS in multi-regional adenomyosis samples with those in adjacent NE/NM tissues (Fig. 4a–c, Supplementary Figs. 10–12, and Supplementary Data 17–19). Most mutations were detected only in adenomyotic lesions but some, including those of *KRAS* and *PIK3CA*, also appeared in NE (e.g., *SERPINA* in Fig. 4a, *PIK3CA* in Fig. 4b, and *KRAS* in Fig. 4a, c). Thus, these mutations likely arose in NE cells before they invaded the NM, consistent with a previously proposed etiology of endometriosis[21]. However, although adenomyosis frequently co-occurs with endometriosis, to date there has been no molecular evidence to support a common cellular origin. Significantly, we found identical *KRAS* mutations in co-existing adenomyotic and endometriotic lesions in multiple patients (Fig. 4d–i, Supplementary Figs. 13 and 14, and Supplementary Data 20 and 21). These results are in line with the frequent co-occurrence of endometriosis anatomical subtypes, such as EN-OV ($p = 0.001$; Table 1) and EN-DI ($p = 0.0319$; Table 1), in our cohort of *KRAS*-mutated adenomyosis patients (Table 1 and Supplementary Data 22). Our data collectively raise the possibility that *KRAS*-mutated clones arising in NE acquire enhanced invasiveness and proliferative capacity that enable them to grow ectopically, driving adenomyosis.

***KRAS*-mutated clones in normal tissue may lead to adenomyosis.** As noted above, recent reports have revealed frequent *KRAS* and *PIK3CA* mutations in NE, i.e., in uterine endometrium that are histologically normal and unaffected by endometriosis or adenomyosis[23,24]. We also detected these mutations in NE adjacent to adenomyotic lesions, prompting us to investigate the relevance of *KRAS*- and *PIK3CA*-mutated clones to the molecular pathogenesis of adenomyosis. To this end, we examined whether the frequency of these mutations was altered in NE adjacent to an adenomyotic lesion. We collected either fresh-frozen or FFPE samples of uterine tissue from 56 individuals: 18 individuals with adenomyosis (group A), 14 individuals with endometriosis but not adenomyosis (group E), and 24 individuals with neither endometriosis nor adenomyosis (group Non-A/E) (Supplementary Data 23 and 24). After enrichment by macro-

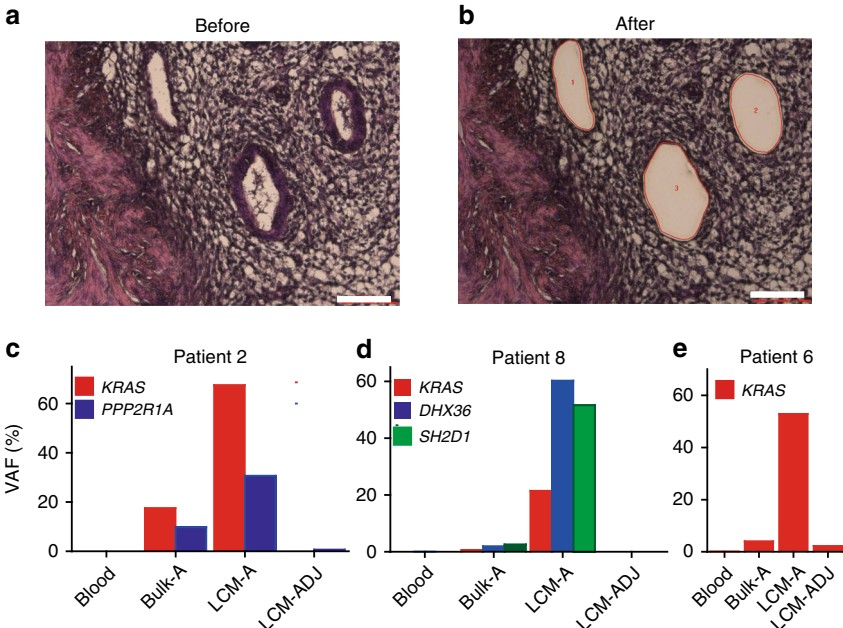

**Fig. 2 Somatic mutations in the epithelial component of adenomyosis lesions. a, b** HE staining of epithelial cells from adenomyosis Patient #6 (**a**) before and (**b**) after isolation by laser capture microdissection (LCM). Scale bars, 100 μm. **c–e** Quantification of VAF as determined by TDS of the indicated genes in peripheral blood (Blood), frozen adenomyosis lesions (Bulk-A), LCM-acquired epithelial component of adenomyosis (LCM-A), or LCM-acquired adjacent muscle cells (LCM-ADJ) from the indicated patients.

dissection, samples of NE and adjacent NM (control) for each individual were subjected to TDS, to assess *KRAS*, *PIK3CA*, and *PPP2R1A* status (Fig. 5). Among macro-dissected NE samples, *KRAS* mutations were commonly observed in group A (10/18 = 55.6%) and group E (7/14 = 50%), but less frequently in group Non-A/E (7/24 = 29.1%) (Fig. 5a and Supplementary Data 24–26) as determined by our mutation criteria (see Methods). Mutations in *PIK3CA* were also observed but to a lesser extent: group A (2/18 = 11.1%), group E (5/14 = 35.7%), and group Non-A/E (6/24 = 25%) (Fig. 5a and Supplementary Data 23–25). These observations are consistent with recent publications showing that recurrent *KRAS* and *PIK3CA* mutations can occur in the endometrium that appears to be histologically normal[23,24]. Next, we compared the VAFs of *KRAS*, *PIK3CA* and *PPP2R1A* hotspot mutations in the macro-dissected NE of group A vs. group E vs. group Non-A/E (Fig. 5b and Supplementary Data 26). Importantly, the VAFs of mutations encoding oncogenic KRAS p. G12/G13 alterations (but no other mutations) were significantly enhanced in group A (mean value: 1.641%) compared with group Non-A/E (0.301%) (*p* = 0.008; Fig. 5b and Supplementary Data 26), suggesting that *KRAS*-mutated clones had expanded in NE of individuals with adenomyosis. We found that *KRAS* mutations were not significantly more frequent in samples from group A (*p* = 0.117) and group E (*p* = 0.297) than in those from group Non-A/E (Fig. 5a). However, our VAF analysis revealed a statistically significant increase in *KRAS*-mutated clones in group A compared with group Non-A/E (*p* = 0.008) (Fig. 5b). Follow-up in a suitably large number of additional patient samples is required, but if corroborated, the latter result suggests our hypothesis that an increase in the frequency of *KRAS*-mutated clones in NE might be the origin of adenomyosis. Taken together, these genomic analyses of NE suggest that an increase in expanded *KRAS*-mutated clones in this tissue may be an early step in the molecular pathogenesis of adenomyosis.

**KRAS mutations reduce DNG efficacy by PR silencing.** Many adenomyotic patients initially receive hormonal treatment,

including administration of the PR agonist DNG[17,25]. Those patients who do not respond to such hormonal treatment then undergo surgery. Our analyses revealed intriguing clinical differences between cases of adenomyosis with *KRAS* mutations vs. those without. Specifically, *KRAS* mutations were more frequent in lesions of patients who had been pretreated with DNG (84% (11/13)) compared with non-pretreated patients (26% (15/57); *p* = 0.0002), a difference not found in patients treated with GnRHa (Table 1 and Supplementary Data 22). This observation generates two hypotheses as follows: (1) DNG may be less efficacious in patients bearing *KRAS*-mutated adenomyotic clones, allowing these abnormal cells to persist until surgery; and (2) DNG treatment drives an enrichment of *KRAS* mutations in adenomyosis. We next sought to distinguish between these two possibilities. DNG reportedly exerts an anti-proliferative effect on cells of the epithelial component of adenomyosis[26]. We examined the effect of mutated *KRAS* expression on DNG's ability to suppress proliferation by taking advantage of an immortalized uterine endometrial epithelial cell line. We engineered these cells to overexpress wild type KRAS (*KRAS*-WT), mutated KRAS (*KRAS*-Mut; G12C or G12D), PIK3CA-WT, or mutated PIK3CA (*PIK3CA*-Mut; E545K or H1047R), and then assessed their viability and proliferation following exposure to DNG (Fig. 6a–c). We observed that a greater percentage of DNG-treated immortalized endometrial cells expressing *KRAS*-Mut were viable compared to empty vector-transfected controls (Fig. 6b). Furthermore, analysis of BrdU incorporation showed that *KRAS*-Mut expression abrogated DNG's ability to suppress cell proliferation (Fig. 6c). In contrast, expression of mutated *PIK3CA* in immortalized cells had no effect on DNG efficacy (Fig. 6b, c). Thus, expression of mutated *KRAS* interferes with DNG's ability to reduce cell proliferation. Taken together, these data support our hypothesis #1 (*KRAS*-mutated adenomyotic clones are less sensitive to DNG treatment) rather than hypothesis #2 (DNG drives *KRAS* mutations in adenomyosis patients). However, it should be noted that future follow-up studies using an appropriately sized cohort will be necessary to reach a definitive conclusion.

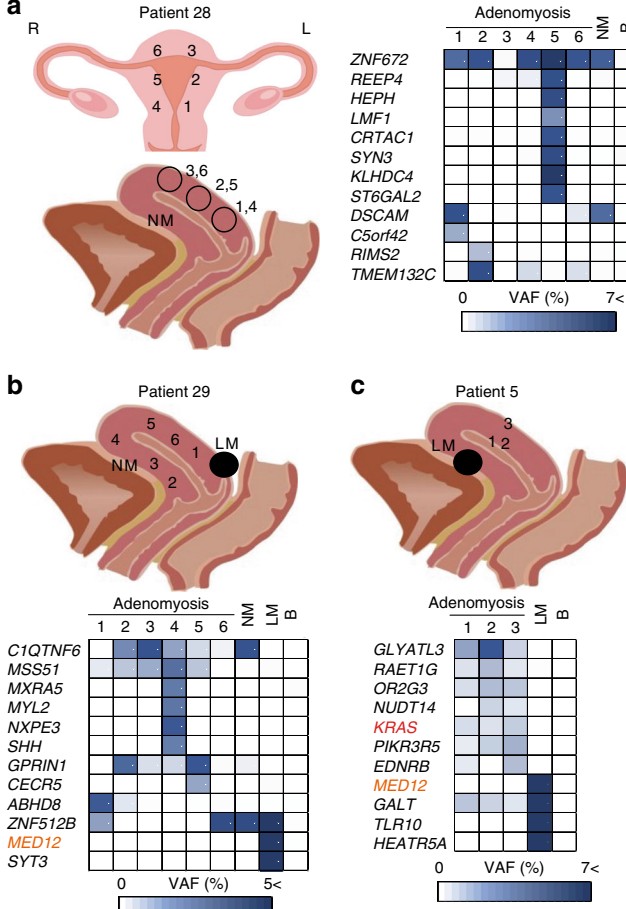

**Fig. 3 Oligoclonality in adenomyosis tissues as revealed by multiregional sampling. a** Left panel: coronal plane (top) and sagittal plane (bottom) schemes indicating the sites of multi-regional adenomyosis samples (1–6) and adjacent histologically normal myometrium (NM) that were acquired from Patient #28 and subjected to mutational profiling. R, right; L, left. Right panel: VAFs of mutations in the indicated genes in the indicated multi-regional samples profiled for each individual adenomyosis lesion, NM, or peripheral blood (B) sample from Patient #28. VAF values are shown using a color scale. **b, c** Top panels: sagittal plane schemes indicating the sites of multi-regional adenomyosis samples plus NM and leiomyoma (LM) samples that were acquired from Patient #29 (**b**) or Patient #5 (**c**) and subjected to mutational profiling. Bottom panels: analysis of VAFs for the samples in the top panels as for **a**. For **a–c**, raw VAF values (%) for each sample are shown in Supplementary Data 14–16.

DNG is a PR agonist, which prompted us to examine PR protein levels in *KRAS*-mutated adenomyosis. We performed immunohistochemical analyses to estimate PR protein levels in the epithelial component of adenomyosis lesions from all 70 of our adenomyosis patients (Fig. 7, Table 1, and Supplementary Data 22). A decrease in PR protein specifically in the epithelial component was significantly more frequent in samples from patients with *KRAS*-Mut (13/26; 50%) than in samples from patients with *KRAS*-WT (7/44; 15.91%) ($p = 0.052$; Table 1). Strikingly, these findings recapitulate those reported for endometrial cancer[27].

There are two major PR isoforms, PR-A and PR-B, which are transcribed from different promoters in a single gene[28]. To investigate whether the reduced PR protein in *KRAS*-mutated adenomyosis samples was due to an effect of mutant KRAS on *PR* mRNA expression, we applied qPCR analysis to our immortalized

endometrial cells overexpressing *KRAS*-Mut, *KRAS*-WT, *PIK3CA*-WT, or *PIK3CA*-Mut (please see the Source Data file). We found that *PR-A/B* mRNAs were indeed downregulated in cells expressing *KRAS*-Mut but not in those expressing *KRAS*-WT, *PIK3CA*-WT, or *PIK3CA*-Mut (Fig. 6d). Although these in vitro overexpression data derived from a single cell line do not show whether *KRAS* mutation downregulates *PR* mRNA directly or indirectly, they suggest that *KRAS*-Mut is linked to downregulation of *PR* expression. Further experiments employing additional independent cell lines and/or patient-derived primary cultures should be conducted in the future to validate these results.

Lastly, we sought to determine how *PR* downregulation by *KRAS*-Mut might be achieved. Expression of *PR-A/B* is known to be regulated, at least in part, by methylation of the *PR* gene promoters[28–30]. In endometriosis, epigenetic silencing of *PR-A/B* imposed by increased methylation of CpG islands within exon 1 has been observed[29]. To investigate whether increased DNA methylation was involved in downregulating *PR* in *KRAS*-Mut adenomyosis, we performed bisulfite sequencing of DNA from the enriched epithelial component of adenomyotic lesions from PR-expressing *KRAS*-WT samples ($n = 20$) and from all 13 PR-negative *KRAS*-Mut samples. In PR-expressing *KRAS*-WT adenomyosis, no significant DNA methylation of CpG islands in the *PR-A/B* promoters was detected (Fig. 8 and Supplementary Data 27). In contrast, there was a marked increase in the methylation level of the *PR-A/B* promoters in *KRAS*-Mut adenomyosis. These results suggest that epigenetic silencing may contribute to the *PR* downregulation observed in *KRAS*-mutated adenomyosis, and that DNG may be less effective in treating *KRAS*-mutated adenomyosis due to this suppression of *PR-A/B* expression.

## Discussion

Our NGS-based study provides fresh insights into the pathogenesis of adenomyosis. We have shown that: (i) *KRAS*-mutated adenomyotic clones originate from NE (Fig. 4a–c); (ii) *KRAS* mutations are shared by co-occurring adenomyosis and endometriosis lesions (Fig. 4d–i); and (iii) the presence of *KRAS* mutations in adenomyosis is associated with the co-occurrence of endometriosis (Table 1 and Supplementary Data 22). Although inflammation caused by ectopic proliferation of epithelial cells and/or de novo metaplasia may play a role in the genetic alterations and disease pathogenesis in some adenomyosis patients, our data suggest that both adenomyotic and endometriotic clones can arise from NE, explaining the high frequency of co-occurrence of these disorders. Nevertheless, we did find evidence of clonal heterogeneity in their development. In Patients #6 and #24, the adenomyotic and endometriotic lesions contained different *KRAS* mutations (Figs. 4f, i, Supplementary Figs. 15 and 16, and Supplementary Data 20 and 28). Non-*KRAS* mutations may also be involved in some cases, as Patient #4 exhibited the same *HIPK1* alteration in her *KRAS*-WT adenomyotic and endometriotic lesions (Supplementary Figs. 15C–D and 17, and Supplementary Data 29).

Recent genomic analyses have shown that histologically NE can bear a relatively high burden of mutations that includes recurrent *PIK3CA* and *KRAS* alterations[23,24]. Consistent with these observations, we also detected recurrent *PIK3CA* and *KRAS* mutations in individuals without adenomyosis or endometriosis (Fig. 5a). However, the frequency and VAF of such mutated clones were significantly elevated in adenomyosis samples (Fig. 5b), suggesting that additional alterations in *KRAS*-mutated clones in the NE might occur very early during the molecular pathogenesis of adenomyosis. Although our study did not definitively identify a

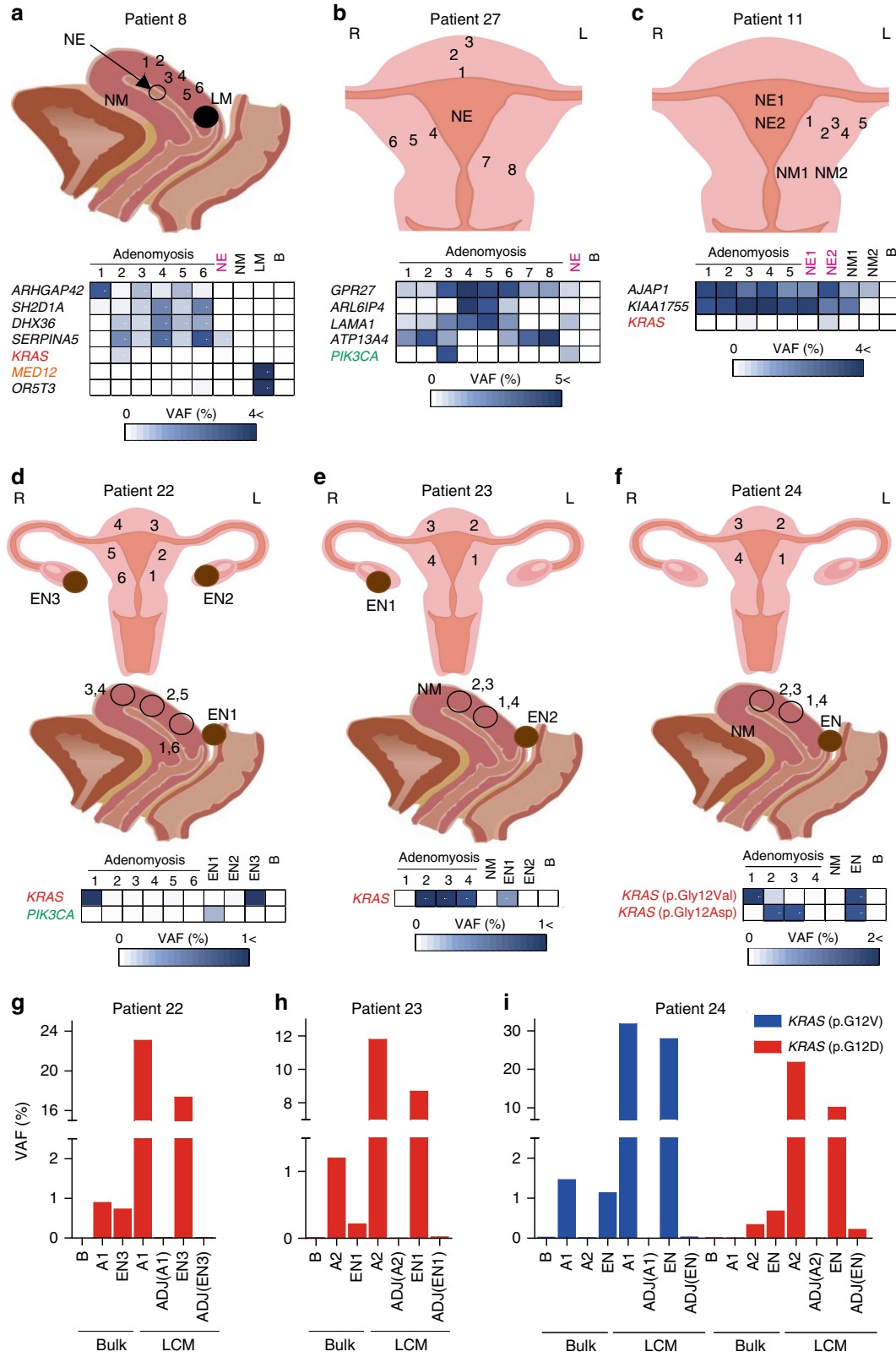

common or first event of genetic and/or epigenetic alteration establishing NE clonality, from a mechanistic point of view, *KRAS* or *PIK3CA* mutation would be considered likely to provide NE cell clones with the ability to proliferate and invade neighboring NM and/or other organs, driving the development of adenomyosis. Mutations of *MED12* and/or *FH*, which are common in

leiomyoma and show high VAFs, have been reported in ~10% of adenomyosis and adenomyotic polyps[31,32]. However, for unknown reasons, these alterations were not detected in our NGS analyses of our adenomyosis cohort (Supplementary Data 6 and 15–17). Lastly, the burdens of total and cancer-associated mutations are known to become elevated in NE by ageing and high

**Fig. 4 Multi-regional sampling reveals shared mutations in adenomyosis, endometriosis, and adjacent normal endometrium.** Sagittal plane scheme (top) and VAF analysis (bottom) of sites of multi-regional adenomyosis (1–6) plus normal endometrium (NE), normal myometrium (NM), leiomyoma (LM), and blood (B) samples from Patient #8. VAF values are depicted as in Fig. 3. **b, c** Coronal plane scheme (top) and VAF analyses (bottom) of multi-regional adenomyosis plus NE and/or NM samples from the indicated patients. **d–f** Coronal plane scheme (top), sagittal plane scheme (middle), and VAF analyses of multi-regional adenomyosis samples plus endometriosis samples (EN) and/or NM samples from the indicated patients. For **a–f**, raw VAF values (%) for each sample are shown in Supplementary Data 17–19. **g–i** Quantification of VAF values as determined by TDS of the indicated genes in frozen (Bulk) and LCM tissues from the indicated patients. Bulk tissues included peripheral blood (B), adenomyotic lesions (A), and endometriotic lesions (EN). LCM tissues included epithelial component of adenomyosis (A), endometriosis (EN), and normal adjacent tissues (ADJ). For **g–i**, raw VAF values (%) for each sample are shown in Supplementary Data 21.

**Table 1 Relationship between patients' characteristics and KRAS mutation status in adenomyosis.**

| Characteristic | KRAS wild-type patient (N = 44) | KRAS-mutated patient (N = 26) |
|---|---|---|
| Median age at operation (range), years | 42.5 (21–50) | 41 (30–58) |
| Preoperative treatment, no. (%) | 25/44 (56.82) | 19/26 (73.08) |
| DNG, no. (%) | 2/44 (4.55) | 11/26 (42.31) |
| GnRHa, no. (%) | 23/44 (52.27) | 9/26 (34.62) |
| LNG-IUS, no. (%) | 2/44 (4.55) | 1/26 (3.85) |
| PR-IHC | | |
| Negative, no. (%) | 7/44 (15.91) | 13/26 (50) |
| Disease co-occurrence | | |
| Endometriosis, no. (%) | 26/44 (59.09) | 22/26 (84.62) |
| EN-OV type, no.(%) | 12/44 (27.27) | 18/26 (69.23) |
| EN-DI type, no.(%) | 9/44 (20.45) | 12/26 (46.15) |
| EN-PE type, no.(%) | 14/44 (31.82) | 3/26 (11.54) |
| Leiomyoma, no. (%) | 28/44 (63.64) | 13/26 (50.00) |
| Ovarian cancer, no. (%) | 0/44 (0.00) | 5/26 (19.23) |
| Endometrial cancer, no. (%) | 0/44 (0.00) | 1/26 (3.85) |
| Cervical cancer, no. (%) | 1/44 (2.27) | 0/26 (0.00) |
| MRI | | |
| Endometrial subtype, no. (%) | 5/44 (11.36) | 1/26 (3.85) |
| Subserous subtype, no. (%) | 9/44 (20.45) | 7/26 (26.92) |
| Others (diffuse), no. (%) | 28/44(63.64) | 18/26 (69.23) |
| Unknown, no. (%) | 2/44 (4.55) | 0/26 (0.00) |
| History | | |
| Caesarean section, no. (%) | 5/44 (11.36) | 0/26 (0.00) |
| Dilatation and curettage, no. (%) | 13/44 (29.55) | 7/26 (26.92) |
| Vaginal delivery, no. (%) | 5/44 (11.36) | 8/26 (30.77) |
| Abortion or stillborn, no. (%) | 17/44 (38.64) | 10/26 (38.46) |
| Premature birth, no. (%) | 1/44 (2.27) | 2/26 (7.69) |
| Hysterectomy, no. (%) | 16/44 (36.36) | 15/26 (57.69) |
| Gravidity, no. (%) | 23/44 (52.27) | 14/26 (53.84) |
| Smoking history, no. (%) | 8/44 (18.18) | 3/26 (11.54) |

*DNG* dienogest, *EN-DI* deep infiltrating endometriosis, *EN-OV* ovarian endometrioma, *EN-PE* peritoneal endometriosis, *GnRHa* gonadotrophin-releasing hormone agonist, *PR-IHC* immunohistochemistry of progesterone receptors shown in Fig. 7, *LNG-IUS* levonorgestrel-releasing intrauterine systems

increases relapse risk[3,6,7]. Our mutational characterization suggests that both of these disorders originate from NE, raising the concern that patients receiving uterus-sparing cytoreductive surgery face a significant relapse risk. Over the past decade, DNG has proven effective for treatment of endometriosis and has been approved for such. However, DNG has also been used for treatment of adenomyosis-related symptoms[26,30], despite a lack of formal knowledge of its side-effects and efficacy in this context. Indeed, efficacy of DNG in treating adenomyosis reportedly varies by individual, in contrast to treatment with GnRHa[34,35]. Although DNG potentially has the advantages of relatively long-term efficacy and oral administration, there is currently no proper therapeutic management protocol for its use in treating adenomyosis[16]. Our work using an immortalized cell system in vitro has suggested that adenomyotic lesions frequently contain *KRAS* mutations that may epigenetically downregulate *PR* and thus reduce DNG efficacy. This observation should be followed up in a large number of DNG-treated adenomyosis patients to determine if our hypothesis is valid and if its implications are translatable to the clinic.

In our cohort of adenomyosis patients, pretreatment with LNG-IUS, an intrauterine device producing local progestin, showed no outcome correlation with the presence of *KRAS* mutations (Table 1). We speculate that this discrepancy could be due to differences in the effective concentration of progestin delivered to the uterus by local (LNG-IUS) vs. oral (DNG) administration. It may also be relevant that only three LNG-IUS patients were available for our analysis. In any case, at least for oral DNG therapy, *KRAS* status may be a biomarker of treatment efficacy, and testing a needle biopsy of an adenomyosis lesion for *KRAS* mutation and/or PR expression might be a valuable diagnostic option. Obviously, further investigations are required to establish the validity of this approach, but investigation of the *KRAS* and *PIK3CA* mutation status of non-diseased uterine tissue of adenomyosis patients might greatly aid in surgical decision-making and therapeutic management.

## Methods

**Ethics and sample collection.** Patients with adenomyosis gave written informed consent prior to their participation in this study. This project was approved by the institutional ethics committees of the University of Tokyo (Project Number G10035), the Juntendo University Faculty of Medicine (Project Number 2014176), and the National Cancer Center Research Institute (Project Number 2015–202). The establishment of this cell line was approved by the Ethics Committees of the National Cancer Center and Nagoya University, and the subject gave informed consent for the use of her clinical samples. Tissue samples from patients with adenomyosis were obtained at the University of Tokyo Hospital and Juntendo University Hospital between December 2016 and July 2019. All patients underwent magnetic resonance imaging for diagnostic purposes. Adjacent normal tissues, which were grossly and tactically distinguishable from adenomyotic lesions, were collected during surgery. Histological review by the study pathologist confirmed that there was no significant contamination of normal tissue by adenomyotic cells. See also Supplementary Data 1 and 2, and 22–24 for additional clinical details pertaining to these specimens.

With respect to therapies, prior to surgery, patients with adenomyosis underwent treatment with GnRHa, DNG, or LNG-IUS according to their

body mass index[24]. Although age did not correlate with frequency of *KRAS* mutations in our cohort (Supplementary Data 25), it would be intriguing to determine whether these factors are also relevant in the contexts of adenomyosis and endometriosis, a question demanding future studies using a larger patient cohort.

Relapses of endometriosis and adenomyosis are relatively common[6,33] and co-existence of adenomyosis and endometriosis

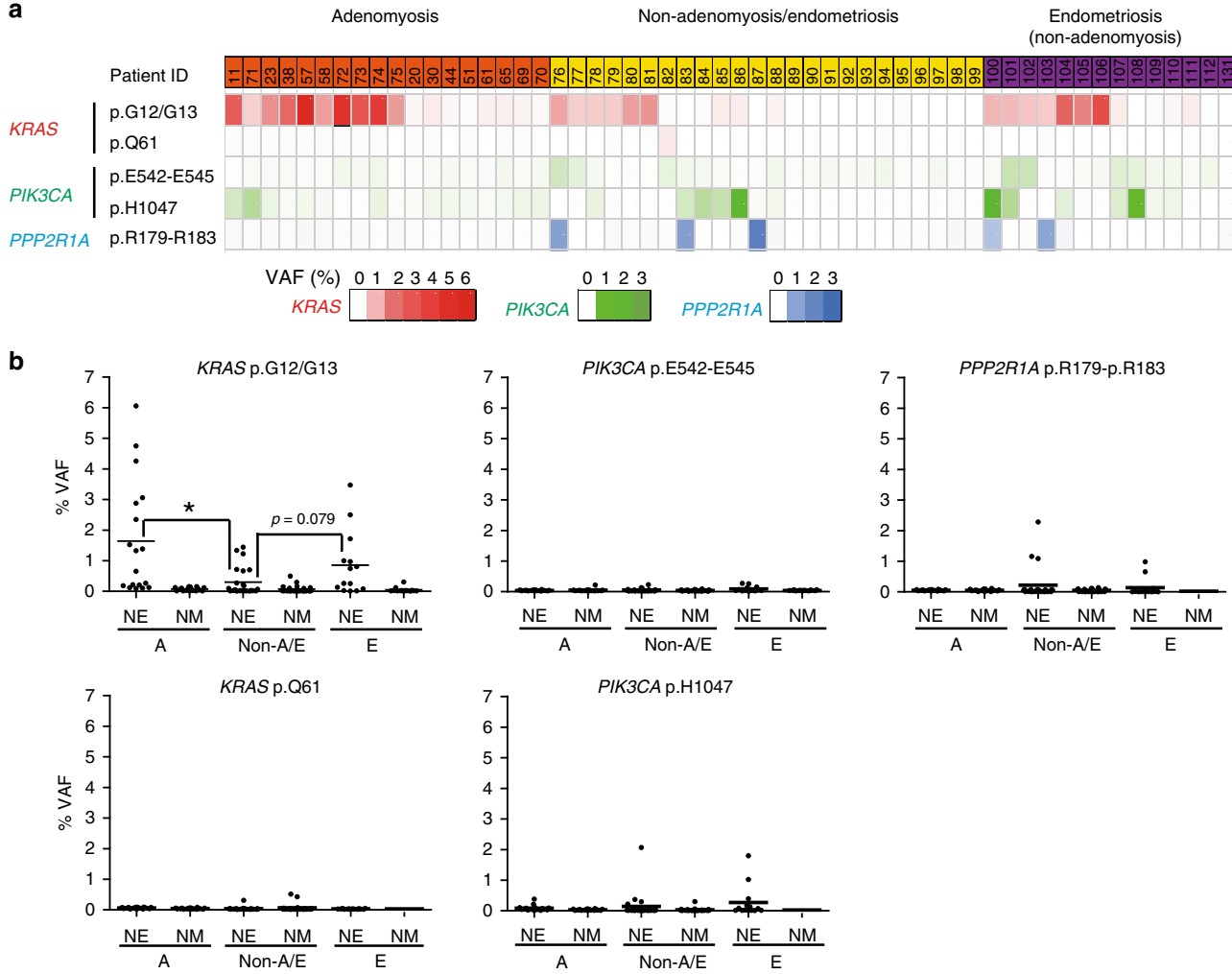

**Fig. 5 Increased mutation frequency and VAF of *KRAS*-mutated clones in the histologically normal endometrium adjacent to adenomyotic or endometriotic lesions. a**, **b** Histologically normal myometrium (NM; negative control) and endometrium (NE) were macro-dissected from the indicated three groups of patients. **a** Occurrence of the indicated gene mutations in NE of patients with adenomyosis (patient ID column: orange), patients with neither adenomyosis nor endometriosis (patient ID column: yellow), and patients with endometriosis but without co-occurring adenomyosis (patient ID column: purple). Bottom boxes: each box represents an individual mutation of *KRAS* (red), *PIK3CA* (turquoise), or *PPP2R1A* (blue). VAF values are shown using individual color scales. **b** Quantification of VAF values as determined by TDS of the indicated mutations in the macro-dissected NM (negative control) or NE from the patients in the group A (adenomyosis; orange in **a**), group Non-A/E (patients with neither adenomyosis nor endometriosis; yellow in **a**), and group E (patients with endometriosis but without co-occurring adenomyosis; purple in **a**). Source data are provided as a Source Data file. Data are values for individual patients plus the mean. *$p < 0.05$ by Welch's *t*-test.

symptoms. GnRHa was mainly used to improve blood hemoglobin levels in patients with severe anemia due to heavy menstrual bleeding (HMB). For GnRHa treatment, subcutaneous injection of 1.88 mg leuprorelin acetate or 1.8 mg goserelin acetate was initiated during menstruation and repeated every 4 weeks. DNG was used to relieve symptoms in patients with dysmenorrhea alone, whereas LNG-IUS was used for those with dysmenorrhea and HMB. For DNG, daily oral administration of 2 mg was started on days 2–5 of menstruation. For LNG-IUS, 52 mg LNG was inserted into the uterine cavity on day 7 of menstruation.

**Next-generation sequencing analyses**. Lesions and adjacent normal tissues were obtained from surgically resected specimens, immediately frozen after resection (fresh-frozen), and stored at −80 °C. Samples of peripheral blood and/or mono-cytes from ascites fluid were stored at −30 °C. See Supplementary Data 3 for additional sample information.

Genomic DNA was extracted from specimens using the QIAamp DNA Blood Midi Kit or the QIAamp Fast DNA Tissue Kit (Qiagen) and quantified by Nanodrop (Thermo Fisher). Fragmentation of genomic DNA was performed using a Covaris LE220 sonicator to produce fragments of ~200 bp and sequencing libraries were generated using the NEBNext Ultra DNA Library Prep kit for Illumina (New England BioLabs). Exonic fragments were enriched with the SureSelect Human All Exon Kit v6 (Agilent Technologies) and paired-end libraries were analyzed and quantified using 2200Tapestation (Agilent Technologies) and a Qubit Fluorometer (Invitrogen). Massively parallel, paired-end sequencing of sample libraries was performed with a HiSeq2500 sequencer (Illumina) as previously described[36] (Supplementary Data 4).

WES reads were independently aligned to the human reference genome (hg38) using BWA, Bowtie2 (http://bowtie-bio.sourceforge.net/bowtie2/index.shtml), and NovoAlign (http://www.novocraft.com/products/novoalign/). Both somatic synonymous and non-synonymous mutations were called using three publicly available mutation callers: MuTect (http://www.broadinstitute.org/cancer/cga/mutect), SomaticIndelDetector (http://www.broadinstitute.org/cancer/cga/node/87), and VarScan (http://varscan.sourceforge.net). Mutations were discarded if any of the following criteria were met: the total read number was <100, the mutant read number was <7, the VAF in disease samples was <0.024, the VAF in the germline control samples was >0.01, the mutation occurred in only one strand of the genome, or the variant was present in normal human genomes in either the 1000 Genomes Project dataset (http://www.internationalgenome.org/) or our in-house database. Gene mutations were annotated by SnpEff (http://snpeff.sourceforge.net). To this end, genomic DNA samples from adenomyosis tissues and corresponding germline controls were subjected to TDS (see details below). Over 100 targets were evaluated in this validation experiment, allowing us to establish criteria that

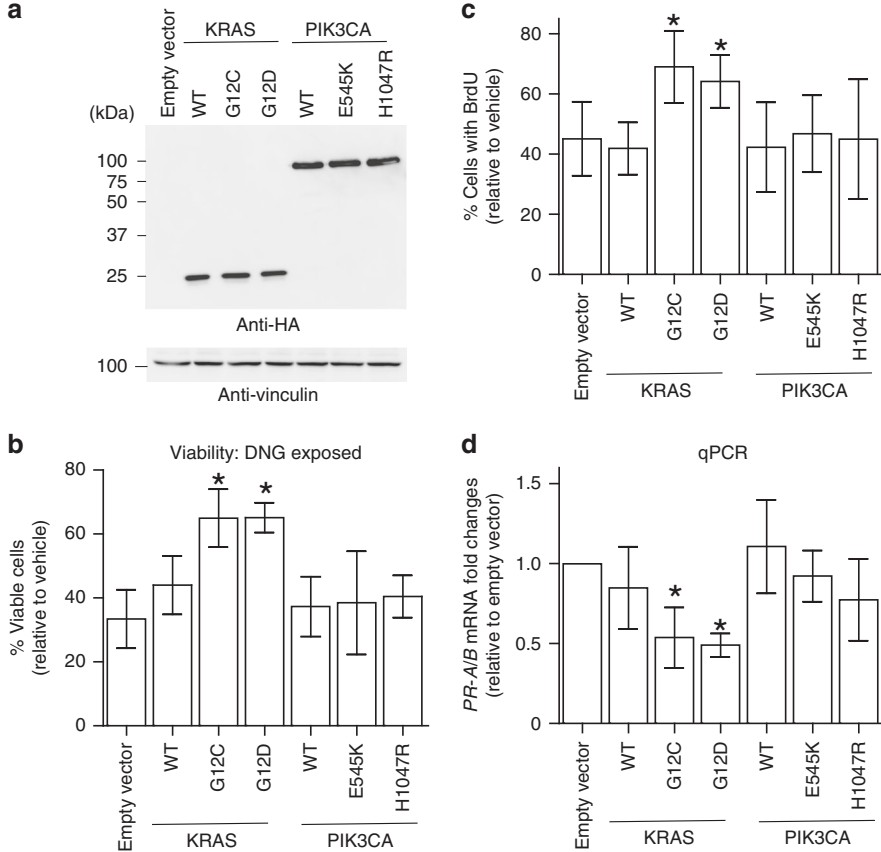

**Fig. 6 Mutant *KRAS* expression abrogates the progestin-induced anti-proliferative effect of DNG through downregulation of *PR*. a** Immunoblotting to detect ectopically expressed, HA-tagged wild-type (WT) KRAS and PIK3CA, or KRAS or PIK3CA bearing the indicated amino acid substitutions, in immortalized human uterine epithelial cells. Vinculin, loading control. Viability (**b**) and proliferation (**c**) of WT immortalized uterine endometrial epithelial cells or those overexpressing the indicated mutations of *KRAS* or *PIK3CA*. **b** Cells were exposed to 1 μM DNG or vehicle (DMSO) for 72 h. Results are the percentage of viable cells relative to vehicle control cells. Data are the mean + SD of six independent experiments, each with three technical replicates per group. *$p < 0.05$ by Welch's *t*-test. **c** Cells were exposed to 1 μM DNG or vehicle (DMSO) for 48 h. Results are the percentage of BrdU+ cells relative to vehicle control cells. Data are the mean + SD of four independent experiments per group without technical replicates. *$p < 0.05$ by Welch's *t*-test. **d** Quantitative RT–PCR determination of mRNA levels of *PR-A/B* in the immortalized cells in **a**. Values were normalized to *GAPDH*. Data are the mean fold change + SD relative to levels in control cells stably infected with empty vector. Six independent experiments, each with three technical replicates per group, were conducted. *$p < 0.05$ by Welch's *t*-test. Source data are provided as a Source Data file.

eliminated 95.4% of validated false-positive calls at the expense of only 15.4% of validated true positive calls (Supplementary Fig. 3).

**Validation of single-nucleotide variations by TDS.** Genomic regions (~250 bp) containing candidate mutated bases were PCR-amplified using appropriate primer sets (Supplementary Data 5). PCR products were purified using AMPure beads (Beckman) and subjected to library construction using the NEBNext Ultra DNA Library Prep Kit for Illumina (New England BioLabs). Libraries were sequenced on an Illumina Miseq using Reagent Kit V2 (300 cycles) to generate 150 bp paired-end reads.

As the VAF values of the observed mutations in adenomyosis were quite low, we performed a statistical calculation to determine whether these alterations were true somatic mutations or NGS noise. In general, it is expected that the distribution of the difference in VAFs between lesion and control samples follows a normal distribution. For each position $i$, a difference of VAFs between adjacent ($a$) and normal ($n$) is defined by $d_i = \max_{y \in \Sigma} |a_{y,i} - n_{y,i}|$, where $\Sigma$ is the set of nucleotides $\{A, C, G, T\}$. Let $\mu_j$ and $\sigma_j$ be the mean (shown in column M in Supplementary Data 12 and column L in Supplementary Data 14–20 and 28–29) and the SD (shown in column N in Supplementary Data 12 and column M in Supplementary Data 14–20 and 28–29), respectively, of the set $\{d_k | k = j - K, \dots, j - 1, j + 1, \dots, j + K\}$ for the estimated position $j$. If one assumes that $X \sim N(\mu_j, \sigma_j^2)$, then the somatic mutation rate for $j$ is defined by $P(X > d_j) < 0.05$, $d_j > 0.001$ and $n_{y,j} < 0.01$, where $y \in \Sigma$. For $K$ values, we used the values shown in length (bp) (column O in Supplementary Data 12 and column N in Supplementary Data 14–20 and 28–29 for the individual mutation analyses). For assessment of statistical significance of somatic mutations, the $p$-value

of each mutation is shown in column P in Supplementary Data 12 and column O in Supplementary Data 14–20 and 28–29.

**Laser capture microdissection.** FFPE tissue samples were sectioned at 8–10 μm thickness, mounted on polyethylene naphthalate membrane-coated slides (Leica Microsystems), and stained with hematoxylin (Sakura Fineteck) for 1 min and eosin (Sakura Fineteck) for 30 s. The epithelial components of adenomyosis and adjacent normal muscle tissues were separately micro-dissected using an LMD 7000 instrument (Leica Microsystems) as directed by the study pathologist (T.H.S.). Genomic DNA was extracted using the QIAamp DNA FFPE Tissue Kit or QIAamp DNA micro Kit (Qiagen) and subjected to PCR, to amplify candidate mutations using the primers listed in Supplementary Data 5. For direct sequencing, PCR products were purified using a PCR purification kit (Qiagen) and sequenced using an Applied Biosystems 3130 Genetic Analyzer (Applied Biosystems). For TDS, PCR products were purified using AMPure beads and subjected to library construction using the NEBNext Ultra DNA Library Prep Kit (New England BioLabs). Paired-end sequencing (150 bp) was conducted on an Illumina Miseq instrument using Reagent Kit V2 (300 cycles).

**TDS analyses of macro-dissected endometrium and myometrium.** Fresh-frozen samples were embedded in OCT compounds (Sakura), sectioned at 10 μm thickness using a Cryostat CM3000 (Leica Microsystems), and mounted on slide glasses (Matsunami Glass IND. Ltd). FFPE tissue samples were sectioned at 10 μm thickness and mounted on the slide glasses. After hematoxylin and eosin staining, NM and epithelial component of histologically normal NE were macro-dissected under microscope. Genomic DNA was extracted using GeneRead DNA FFPE Kit

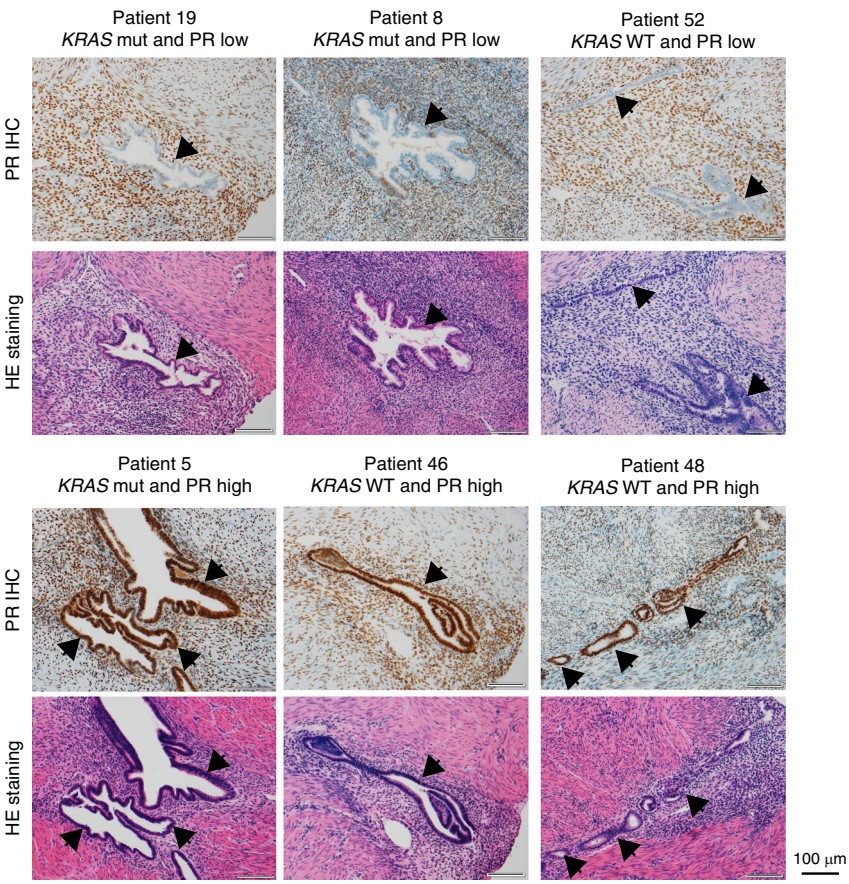

**Fig. 7 Downregulation of progesterone receptors (PRs) in *KRAS*-mutated adenomyosis as assessed by immunohistochemical (IHC) analyses.**
Representative photomicrographs of (upper) IHC of PR and (lower) HE staining of adenomyotic lesions from the indicated patients. Arrowheads, epithelial component. Scale bars, 100 μm.

(Qiagen) and quantified by Qubit fluorometer (Thermo Fisher). TDS analyses were described above. See Supplementary Data 23 for patient and sample information. As peripheral blood samples were not available for the FFPE tissue samples, we had to use histologically normal (NM) tissue samples as a negative control. Most SNVs, including *KRAS* and *PIK3CA* mutations, were only detected in NE but not NM (Fig. 2–4 and Supplementary Data 14–21). As the VAFs of the mutations in the macro-dissected epithelial component of NE were low, we considered an alteration in a sample as a true mutation if its VAF was higher than the mean VAF + 3 SD in the 56 NE samples (Supplementary Data 26). The cutoffs for the SNV are as follows: 0.335% (mean: 0.065%, SD: 0.09%) for KRAS p.G12/G13, 0.302% (mean: 0.05%, SD: 0.084%) for KRAS p.Q61, 0.127% (mean: 0.043%, SD: 0.028%) for PIK3CA p.E542E545, 0.158% (mean: 0.032%, SD: 0.042%) for PIK3CA p. H1047, and 0.15% (mean: 0.042%, SD: 0.036%) for PPP2R1A p.R179-R183.

**Immunohistochemistry.** Adenomyosis samples were fixed in 4% paraformaldehyde (Pierce) at 4 °C for 24 h. Paraffin sections (5 μm) were subjected to immunostaining with anti-progesterone receptor antibody (undiluted; clone: 1E2; Ventana) using the Benchmark XT automated staining system (Roche). Scoring of all PR immunostaining was performed blind by T.H.Y. to avoid evaluator bias.

**Bisulfite sequencing.** Genomic DNA from LCM-enriched adenomyotic lesions was subjected to bisulfite conversion using an EpiTect Bisulfite Kit (Qiagen). Converted DNA samples were amplified by PCR using a Kapa HiFi Uracil+ Kit (Kapa Biosystems). All procedures were performed according to the manufacturer's instructions. The bisulfite converted genomic DNA were subjected to the PCR; 98 ° C for 5 min followed by 50 cycles of 98 °C for 10 s, 60 °C for 10 s, and 72 for 30 s. The primer sequences used for PCR were as follows[6,37].
*PR-A* forward, 5′- GGTTTTGTTAGGGATAGGATTTTTT-3′;
*PR-A* reverse, 5′-ACTACCTCCAACACCCCTTATAACT-3′;
*PR-B* forward-1, 5′-TGTGGGTGGTATTTTTAATGAGA -3′;
*PR-B* reverse-1, 5′-CCCCCTCACTAAAACCCTAAA-3′;
*PR-B* forward-2, 5′-AGTATGGAGTTAGTAGAAGTT-3′;
*PR-B* reverse-2, 5′-TCACAAGTCCAACACTTAAATAACT-3′.

PCR products were purified using AMPure beads (Beckman) and subjected to library construction using the NEBNext Ultra DNA Library Prep Kit for Illumina (New England BioLabs). Libraries were sequenced on an Illumina Miseq using Reagent Kit V2 (300 cycles)(Illumina) to generate 150 bp paired-end reads.

**Immortalized epithelial endometrial cells.** To generate immortalized human endometrial cells, normal endometrial cells obtained from a patient (41 years old) with cervical squamous cell carcinoma were cultivated in Advanced DMEM: F12 (Invitrogen) containing 100 ng/ml EGF (Sigma), 10 μM Y-27632 (Selleck Chemcals), 2% B-27 (Thermo Fisher Scientific), and 10 nM 17β-estradiol (Cal-biochem). Lentiviral vectors expressing TERT, cyclin D1, and mutant CDK4 (CDK4$^{R24C}$: an inhibitor-resistant form of CDK4) were constructed by recombination with a lentiviral vector, CSII-CMV-RfA (a kind gift from Dr. Hiroyuki Miyoshi). To obtain infectious lentiviral particles, 293T cells (ATCC) were transfected with indicated constructs and packaging plasmids (pCAG-HIVgp and pCMV-VSV-G-RSV-Rev) (kind gifts from Dr Hiroyuki Miyoshi). For immortalization, the normal endometrial cells were transduced with mutant CDK4 (CDK4$^{R24C}$), cyclin D1, and TERT via lentivirus-mediated gene transfer. The immortalized cells were seeded in the matrigel (Rensselaer)-coated six-well plates. Human embryonic kidney 293T cells (ATCC) were cultured in DMEM-F12 containing 10% fetal calf serum (Life Technologies). The 293T cells ($5 \times 10^6$ cells) were plated in a 10 cm dish, incubated for 24 h, and then transfected with 1 μg of the retroviral plasmids (HA-tagged wild-type or mutant *KRAS* (G12C or G12D) (Addgene; catalog number 75282, 58901, and 58902) or *PIK3CA* (cloned cDNA subcloned into pBabe backbone vector) with ecotropic packaging plasmids (Takara Bio) for 24 h using Lipofectamine LTX (Thermo Fisher). After 24 h, the virus-containing medium was filtered (0.45 μm filter, Millipore) and supplemented with 4 μg/ml polybrene (Nacalai). For retroviral gene transductions, the culture medium was replaced by the appropriate retroviral supernatant (1:4 dilutions) and incubated at 37 °C. Transduced immortalized cells were cultured with DNG (Sigma) at 37 °C, followed by counting of cell numbers using a Countess II cell counter (Invitrogen). For cell cycle analyses, BrdU incorporation was performed using the APC-BrdU kit (BD Pharmingen) and measured by FACs Canto II (BD Biosciences).

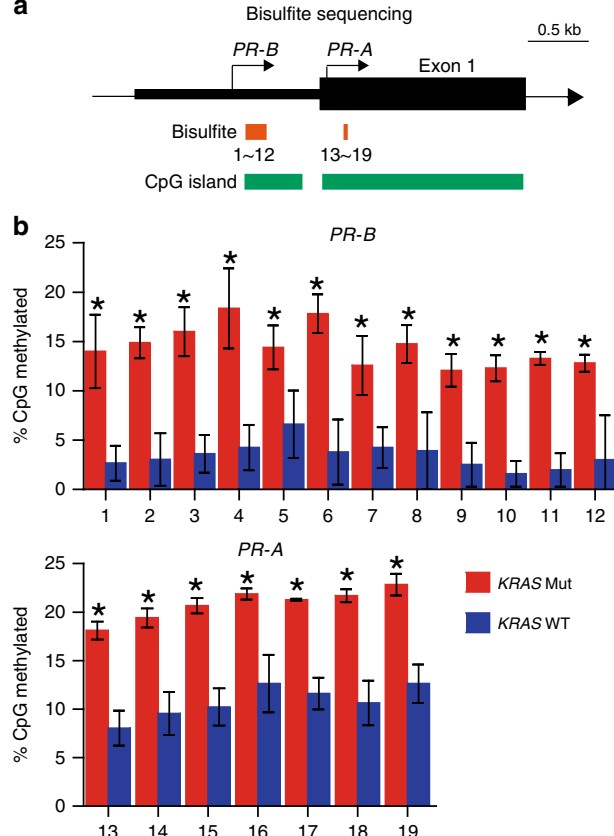

**Fig. 8 Mutant *KRAS* expression is linked to epigenetic silencing of *PR*. a** Schematic diagram of the *PR-A/B* promoters and exon 1. Transcription start sites (arrows), CpG islands, and regions of bisulfite sequencing are indicated. **b** Frequency of methylated CpG in *PR-B* (top) and *PR-A* (bottom) as determined by bisulfite sequencing of DNA from *KRAS*-WT ($n = 20$) and *KRAS*-mutated ($n = 11$) adenomyosis samples. Data are the mean + SD. *$p < 0.05$ by Welch's *t*-test.

**Immunoblotting**. Cells were lysed in 1% Triton X-100 lysis buffer (20 mM Tris-HCl at pH 8, 100 mM NaCl, 10% glycerol) for 30 min on ice. Cell lysates (20–100 µg) were subjected to 10% SDS-polyacrylamide gel electrophoresis and immuno-blotting using anti-vinculin antibody (1:5000: clone SPM227; Abcam) or anti-HA tag antibody (1:1000; clone 6E2; Cell Signaling Technology). Secondary antibody used was horseradish peroxidase-conjugated anti-mouse immunoglobulin G (1:5000; NA931V; GE Healthcare).

**Real-time RT–PCR**. Total RNA was extracted using the RNeasy Mini kit (Qiagen) and reverse-transcribed using the SuperScript IV VILO cDNA synthesis kit VI (Thermo Fisher). The resulting cDNAs were diluted to 1:5–10 and served as templates for real-time PCR using Power SYBR Green PCR Master Mix (Applied Biosystems) plus 250 nM forward primer and 250 nM reverse primer on an ABI 7700 instrument (Applied Biosystems). All procedures were performed according to the manufacturer's instructions. The primer sequences used for real-time reverse transcriptase–PCR were as follows; *GAPDH* forward, 5′-GGAAGCTCACTGGC ATGGCC-3′; reverse, 5′-CCTGCTTCACCACCTTCTTG-3′; *PR-A/B* forward, 5′-GAGCACTGGATGCTGTTGCT-3′; reverse, 5′-GGCTTAGGGCTTGGCTTT C-3′; *KRAS* forward, 5′-GCAAGAGTGCCTTGACGATAC-3′; reverse, 5′-TC CAAGAGACAGGTTTCTCCA-3′; *PIK3CA* forward, 5′-TCTGAACGTTTGTA AAGAAGC-3′; reverse, 5′-CATTATTTGGAGAAACTATTA-3′. Data were nor-malized to the housekeeping gene *GAPDH*. Results were calculated using the comparative threshold cycle method ($2^{-\Delta\Delta Ct}$).

**Statistics**. Statistical analyses used for judging somatic mutations were described in the Methods section under the heading "Validation of single-nucleotide varia-tions by TDS". Statistical significance values shown in Figs. 5b, 6b–d, and 8 were determined using Welch's *t*-test. Comparisons of the distribution of categorical variables in different groups were performed using Fisher's exact test (Table 1).

**Reporting summary**. Further information on research design is available in the Nature Research Reporting Summary linked to this article.

## Data availability

The whole exome sequencing data have been deposited in the National Bioscience Database Center (NBDC) under the accession code JGAS00000000169. All the other data supporting the finding this study are available within the article, Supplementary Information file, Supplementary Data file, or Source Data file and from the corresponding author upon reasonable request. The source data underlying Figs. 5b and 6 are provided as a Source Data file. A reporting summary for this article is available as a Supplementary Information file.

## Code availability

In this study, we extensively used publicly available algorithms and did not use any custom code.

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

## Acknowledgements

We thank members of Dr Mano's lab, particularly Dr Shinji Kohsaka and Ms Takako Matsumoto, and National Cancer Center Research Institute Core Facility, particularly Ms Yuriya Shiotani and Mr Naoaki Uchiya, for their expert technical support. We are grateful to Drs Kelsie Thu and David Cescon for their constructive suggestions. We thank Ms. Chiho Kohno for her technical assistance and Dr. Hiroyuki Miyoshi (RIKEN, BioResource Center; presently Keio University) for providing plasmids. We also thank Dr Momoe Itsumi for her cooperation in creating original illustrations showing in Figs. 3 and 4, and Supplementary Figs. 13D and 15A, C. We are also grateful to all the women who have generously donated their tissues to be used in our research studies; without them, this work would not have been possible. This study was supported in part by JSPS KAKENHI grants (#19K07708 to S.I., #19H03144 to Y.H., #17K07250 to T.U., #19H03796 to Y.O., and #18K09299 to Y.T.E.); a grant for Leading Advanced Projects for Medical Innovation (LEAP) (#JP17am0001001h0004 to H.M.); Project for Cancer Research and Therapeutic Evolution (P-CREATE) (#JP19cm0106502 to M.K.), and Project for "Whole Implementation to Support and Ensure the female life" (WISE) (#JP19gk0210021h0001 to Y.H.) from the Japanese Agency for Medical Research and Development; and a grant from Suzuken Memorial Foundation (to S.I.).

## Author contributions

S.I., Y.H., K.O., Y.O., Y.T.E., M.K., and H.M. conceived the project and designed the study. Y.H., Y.F., E.Y., A.T., T.T., and Y.T.E. collected clinical samples. S.I. and R.T. performed NGS experiments. T.U. and S.K. performed NGS data analyses. T.H.Y., T.H.S., M.I., and T.S. performed histological analyses. T.K., A.M., and S.O. provided immortalized endometrial cells. Y.T.A., S.S., and K.T. provided experimental and analytical support. S.I. and H.M. wrote and edited the manuscript with feedback from all authors.

## Competing interests

The authors declare no competing interests.
