## [Peer Review File · Nature Communications]

Reviewers' comments:

Reviewer #1 (Remarks to the Author):

This is an interesting and highly novel paper describing a somatic mutational genomic analysis of adenomyosis. The authors profiled adenomyosis tissue samples from 70 women (192 multi-regional samples) through whole exome next generation sequencing (mainly), comparing with 'adjacent normal' myometrium and endometrium, as well as tissue samples from co-occurring endometriosis, leiomyoma, and ovarian cancer. They observe a relatively high frequency of KRAS mutations (37%) among adenomyosis, mirroring similar mutation proportions found for endometriosis, and demonstrate through targeted sequencing of epithelial cells isolated through laser capture microdissection that these mutations are concentrated in epithelial cells. They observed an association between KRAS mutation frequency and dienogest treatment prior to surgery, which they suggest implies a potential lack of effectiveness of the progesterone receptor agonist dienogest in KRAS mutation positive adenomyosis. They then investigate the effect of dienogest on proliferation of immortalised endometrial epithelial cell lines, showing that its anti-proliferative effect is diminished in KRAS mutation positive cells. They show through qPCR analysis that PR-A/B gene expression is downregulated in KRAS mutation positive cells; and through bisulfite sequencing of the epithelial component of adenomyosis that the PR-A/B promoters are hyper-methylated, suggesting that epigenetic silencing may be linked to the PR downregulation in KRAS mutation positive adenomyosis. Lastly, they conclude that although these findings need to be further tested, that KRAS mutation profiling of adenomyosis could be a patient stratifier for treatment with dienogest.

This is a very clearly written paper, for which the authors should be commended. It represents a significant addition to the field, in particular for adenomyosis, and also has consequences for the way in which we think about endometriosis.

Main comments

This study only concerns samples from women with adenomyosis. Myometrium, endometrium, or indeed endometriosis from women without adenomyosis were not profiled. This is a significant shortcoming. The mutational frequency in these tissues per se, remains unknown, as is the role in pathogenesis/disease origin. Recent papers have shown that the mutational burden is high in endometrium, likely due to this being a highly dynamic tissue: there is a high frequency of somatic mutation burden in 'normal' tissues, including endometrium; see e.g. doi:

<https://doi.org/10.1101/561050>

and doi: <https://doi.org/10.1101/505685>). This should be discussed.

Table S6 shows a long list of 134 SNVs detected in adenomyosis tissues, but it does not give an indication of their frequency. Without any discussion of these SNVs, the KRAS, PIK3CA, and PPP2R1A mutations are selected in presentation of Fig 1 and the text. Were these the most common? In this context, Fig 1 is also unclear. The number of SNVs are presented on the y-axis in Fig 1C for each patient ID, with mutation status for the three above genes underneath. However, there are patient IDs with 0 SNVs (no grey squares), which appear to have KRAS mutations. Also, page 8 line 3 states that 134 SNVs were detected in 31/51 adenomyosis cases, but should this not be 31/70 (Fig 1)? This means that the majority of adenomyosis samples actually did not carry a mutation.

Page 10, lines 7-12: the authors suggest that, because the mutational signatures detected in adenomyosis 'most closely resemble COMIC signatures associated with ageing' that this suggested adenomyosis is an age-associated disease. Did the mutational frequency/burden in their dataset associate with age?

Page 12, lines 11-13. 'Our data collectively imply that KRAS-mutated clones arising in normal

endometrium acquire enhanced invasiveness and proliferative capacity that enables them to grow ectopically, driving both adenomyosis and endometriosis'. This cannot be concluded, as the mutational frequency of endometrium in non-adenomyosis/non-endometriosis women was not investigated.

Minor comments

Page 4 line 9 states that differences exist in pathogenesis between adenomyosis and endometriosis, whereas page 5 line 1-3 states that a common molecular mechanism is hypothesised. This reads as contrasting information.

Page 6, line 1. When referring to the published mutational results for endometriosis, it is important to add a word of caution that the extent to which these are related to disease mechanism is unknown (see earlier comment re. high frequency of somatic mutation burden in 'normal' tissues, including endometrium)

Page 7, line 4-6: a reference should be made to the methods section for description of clinical sampling. It should be made clear if all patients underwent MRI for diagnostic purposes, and also how 'adjacent normal' tissues were confirmed to be normal: through histology? How was 'adjacent' defined/chosen?

Page 9, line 3: Fig 1G is referenced but does not exist.

Page 13, line 3: 'KRAS mutations were more frequent in patients who have been pre-treated with the PR agonist DNG... raising the possibility that DNG is less efficacious in KRAS-mutated disease'. How this conclusion is made is currently unclear. Presumably, the assumption is that women undergoing surgery for adenomyosis in the present study were doing so, because their treatment was ineffective; i.e. the surgery was an alternative to medication. This should be made clear in the methods. Currently, 'pre-treatment with DNG' to the uninitiated reader could mean a standard treatment prior to surgery, which then begs the question why the above conclusion is made.

Reviewer #2 (Remarks to the Author):

The manuscript by Inoue et al., describes mutational analyses of uterine adenomyosis, a common but benign condition in which ectopic endometrial cells (epithelial and stromal) are present, and grow, within the uterine myometrium. Although adenomyosis is a benign condition, it is associated with chronic pelvic pain, menorrhagia and infertility and is present in ~70-80% of women with endometriosis. Very little is known about the molecular alterations within adenomyosis. Here, Inoue et al., exome sequenced a large series (N=192) of fresh-frozen adenomyosis specimens, as well as specimens of endometriosis (n=15), leiomyoma (n=13), ovarian cancer (n=5), and adjacent normal myometrium (n=13). Peripheral blood or mononuclear cells from ascites served as germline controls. Targeted deep sequencing and data filtration resulted in the identification of 134 unique synonymous and nonsynonymous SNVs in 60.8% of adenomyosis cases, with a mean of 2.6 SNVs/sample. Variant allele frequencies (VAFs) were low (mean 4.8%). VAF was markedly increased in the laser capture microdissected epithelial cell component of adenomyosis, leading them to the conclusion that "adenomyosis may arise from the ectopic proliferation in epithelial cell clones." The KRAS proto-oncogene was mutated in 37.1% (26/70) of adenomyoses, leading to proposal that mutated KRAS may be a pathogenic driver of adenomyosis. Multiregional sampling of adenomyoses from 3 individuals revealed both shared and unique mutations among samples from the same individual, leading to the conclusion that some alterations are acquired early during adenomyosis development while others are more restricted. Most mutations in adenomyosis samples and co-existing leiomyoma differed, indicating that

adenomyosis and leiomyoma are distinct entities. Identical KRAS mutations present in co-existing adenomyotic and endometriotic lesions in multiple patients lead the authors to propose that “KRAS-mutated clones arising in normal endometrium acquire enhanced invasiveness and proliferative capacity that enables them to grow ectopically, driving both adenomyosis and endometriosis.”

To determine whether mutations in adenomyosis originate in histologically normal endometrium or normal myometrium, the study compared mutations detected by targeted deep sequencing of multiregional adenomyosis samples with those identified in the adjacent normal endometrium and the adjacent normal myometrium. The presence of KRAS and PIK3CA mutations in adenomyosis and adjacent normal endometrial cells lead to the conclusion that “these mutations likely arose in normal epithelial cells before they invaded the normal myometrium.”

The authors noted that “KRAS mutations were more frequent in patients who had been pretreated with the PR agonist DNG ($P=0.0002$), raising the possibility that DNG is less efficacious in KRAS-mutated disease.” In subsequent experiments, DNG treatment of immortalized endometrial cells following transduction of constructs expressing wildtype or mutant KRAS showed that “a greater percentage of DNG-treated immortalized endometrial cells expressing KRAS-mutant were viable compared to controls.” The study concludes that expression of mutant KRAS interferes with DNG’s ability to reduce cell proliferation. By immunohistochemistry they showed that decreased PR protein expression in the epithelial component of adenomyosis specimens was significantly more frequent in KRAS-mutated samples (50%) than in KRAS-wildtype samples (16%). By qPCR, they observed downregulation of PR-A/B mRNA expression in immortalized endometrial cells ectopically expressing mutated KRAS versus cells ectopically expressing wildtype KRAS; these findings lead to the suggestion that mutated KRAS directly downregulates PR expression. They further showed a marked increase in the methylation level of the PR-A/B promoters in KRAS-mutated adenomyosis, leading the suggestion that “epigenetic silencing may be responsible for (or contribute to) the PR downregulation observed in KRAS-mutated adenomyosis, and the DNG may be less effective in treating KRAS-mutated adenomyosis due to this suppression of PR-A/B expression.”

Overall, the study concludes that adenomyotic lesions frequently contain KRAS mutations that may epigenetically downregulate PR and thus reduce DNG efficacy. The authors speculate that KRAS status may be a biomarker of treatment efficacy, and testing a needle biopsy of an adenomyotic lesion for KRAS mutation and/or PR expression might be a valuable new diagnostic option. They state that further investigations will be needed to establish the validity of this approach.

Reviewer comments

The molecular analyses of a large series of uterine adenomyosis specimens is valuable and important. However, there are major concerns including concerns with several conclusions drawn from the study findings.

Results section-1 “Somatic mutations are present in adenomyosis”

1) Page 7, line 16 states “Our robust criteria and validation by targeted deep sequencing (TDS) (see Materials and Methods, 1 Figure S3) permitted us to detect 134 unique synonymous and non-synonymous single nucleotide variations (SNVs) in 31/51 (60.8%) adenomyosis cases, defining adenomyosis as a clonal disorder with somatic mutations (Tables S5–6).” Reviewer comment: The conclusion that adenomyosis is a “clonal” disorder is not supported by the aforementioned observation.

2) Page 8, line 6: A mean VAF of 4.8% is noted. The accompanying range should be provided.

Results section-2 “KRAS is recurrently mutated in adenomyosis”

3) Page 8, line 14 states: “We next used the ClinVar database (<https://www.ncbi.nlm.nih.gov/clinvar/>) to select our mutation candidates for further study (see Materials and Methods, Table S10).” Reviewer comment: There is no information describing this methodology in either the “Materials and Methods” or in the “Supplemental Materials and Methods”. This information needs to be provided in order to review the validity of the strategy used to choose candidate mutations from all SNVs and to understand why the authors do not

mention or focus in the recurrent PIK3CA, TP53, and ARID1A mutations listed in Table S12. Also, the authors need to define the meaning of "mutation candidate"; do they mean candidate driver mutation?

4) Page 9, line 2 states "As non-KRAS pathogenic mutations were not linked to adenomyosis using this strategy..." Reviewer comment: What "strategy" are the authors referring to, and what is the intended meaning of "linked"?

5) Table S12: The Tab for this table in the Excel spreadsheet is "Table S12 PTENARID1Adeepseq", but there is no information on PTEN in the table (the table shows data for PIK3CA, TP53, and ARID1A).

Results section-3 "Somatic mutations occur in the epithelial cell component of adenomyosis"

6) The rationale for, and interpretation of, the experiments in this section are disconnected. The rationale states "Only a minor fraction of an adenomyosis lesion is comprised of epithelial cells (Figure 2A). This observation prompted us to consider whether the low VAFs we observed were due to low epithelial component content or limited clonal expansion of mutated cells." After sequencing DNA from laser capture microdissected epithelial and myometrial cells, the study findings and conclusions are "Compared to bulk frozen samples, VAFs were markedly increased in all three LCM-A samples (Figures 2C–E, S6, Table S13), demonstrating that somatic mutation occurs in the epithelial component of adenomyosis. Thus, adenomyosis may arise from the ectopic proliferation of epithelial cell clones". Comment: The authors need to provide an interpretation of their findings in relation to their initial question of whether the low VAFs in unmicrodissected samples were due to low epithelial component content or limited clonal expansion of mutated cells. They also need to state and discuss on the fact that they detected mutations, including KRAS and PPP2R1A mutations, in the laser captured adjacent normal muscle cells (Figure 2C-E).

Results section-4 "Adenomyosis is associated with the mutational signature of aging"

7) It is unclear how meaningful the analysis of mutational signatures is since it is based on an analysis only 148 variants and all variants were pooled and analyzed as a single set (rather than separately assessing the mutational signature of variants in each lesion). In addition, the finding that the pooled mutational signature most closely resemble COSMIC aging signatures is puzzling because the patients in this study are relatively young (range 21yr-58yr; mean 42.5yr for KRAS-wildtype cases; mean 41yr for KRAS-mutated cases) (Table 1). The authors do discuss the latter point.

Results section-5 "Adenomyosis is clonally diverse and does not share mutations with leiomyoma"

8) The authors need to indicate what the variable blue coloring symbolizes in the Tables shown in Figure 3 and 4.

9) Page 11, line 1 states "Some mutations were shared among almost all multi-regional samples from a single patient [e.g : ZNF672 in #28 (Figure 4 3A); C1QTNF and MSS51 in #29 (Figure 3B)], suggesting that these genetic alterations were acquired early during adenomyosis development, while other mutations were more restricted (Figure 3). Most mutations detected in adenomyosis and co-existing leiomyoma were mutually exclusive (Figures 3B–C, 4A, Tables S15–17), implying the lack of a clonal relationship between these disorders. Thus, despite their frequent co-occurrence, adenomyosis and leiomyoma are distinct entities." Comment: The statistical analyses provided for variants in Tables S15, S16 and S17 (column O) seems to indicate that most mutations are not considered somatic. Therefore, it is unclear if the authors are assigning a different interpretation to variants when they are displayed in Tables S15, S16 and S17 versus when they are displayed in the boxes in Figure 3 and Figure 4. These points require clarification in the manuscript.

Results section-7 "KRAS mutations reduce DNG efficacy through epigenetic silencing of PR"

10) The manuscript states "KRAS mutations were more frequent in patients who had been pretreated with the PR agonist DNG17,22 ($p = 0.0002$, Table 1), raising the possibility that DNG is

less efficacious in KRAS-mutated disease." The latter comment is not warranted since the authors do not provide any data on treatment outcome (efficacy) of DNG treatment for their adenomyosis patients. They also do not clarify whether these patients received DNG treatment for their adenomyosis or for co-existing endometriosis. If the treatment was based on co-existent endometriosis this may actually be the feature that is driving the enrichment for KRAS mutations in DNG-treated adenomyosis cases, since this study reports shared KRAS mutations between co-existing adenomyosis and endometriosis.

11) Table 1 shows the frequency of DNG treatment for KRAS wildtype versus KRAS mutated cases. The reader has to use these data to manually calculate the frequency of KRAS mutations in patients pretreated with DNG (11 of 13, 84%) and in untreated patients (15 of 57, 26%).

12) The concentration of DNG used in the cell-based experiments is not provided. The vehicle (DMSO? Other?) in which DNG was resuspended is not provided. For the cell-based assays the comparison seems to be DNG-treated cells versus untreated cells. It should be DNG-treated versus vehicle-treated cells. Until the data on DNG- and vehicle-treatments data are provided the validity of the cell-based assays cannot be determined

13) The Life Sciences study design information states "Data reproducibility was examined by 3 independent experiments in cell viability (Figure 5A)." Were technical replicates included within each of the three experiments?

14) The data displayed in Figure S16 should be included in the main Figure 5 since a major claim of the paper is that mutant KRAS downregulates PR expression. Having said that, I do not find the observed downregulation of PR expression by KRAS-mut (Figure 16B) very convincing. What exactly are the P values for these data? The fold difference in expression seems to shift from ~1.0 to ~0.5.

Supplemental tables

15) There are numerous errors and inconsistencies in the descriptions of Table S1-S12. The authors need to carefully check all tables descriptions for accuracy. As examples:

i) Tab for Table S3 is "Table S3 clinical info", whereas the header for Table S3 is "Table S3. Method used to detect genomic alterations in tissue samples from all patients."

ii) Tab for Table S12 is "Table S12 PTENARID1Adeepseq", whereas the header for Table S12 is "Table S12. Summary of TDS read information for pathogenic mutations defined by the ClinVar database."

Response to Reviewer #1

We thank for the reviewer for his/her general comment: **“This is a very clearly written paper, for which the authors should be commended. It represents a significant addition to the field, in particular for adenomyosis, and also has consequences for the way in which we think about endometriosis.”**

Reviewer #1’s Specific Comments

Major comments

1) **This study only concerns samples from women with adenomyosis. Myometrium, endometrium, or indeed endometriosis from women without adenomyosis were not profiled. This is a significant shortcoming. The mutational frequency in these tissues per se, remains unknown, as is the role in pathogenesis/disease origin. Recent papers have shown that the mutational burden is high in endometrium, likely due to this being a highly dynamic tissue: there is a high frequency of somatic mutation burden in ‘normal’ tissues, including endometrium; see e.g. doi: <https://doi.org/10.1101/561050> and doi: <https://doi.org/10.1101/505685>). This should be discussed.**

We thank the reviewer for this solid and constructive suggestion based on recent publications of genomic analyses of normal endometrium. We agree with the reviewer that our study lacks mutational analyses of normal endometrium (NE) and endometriosis from adenomyosis-free women. As the reviewer noted, two recent online preprint repository reports (bioRxiv) unveiled relatively frequent somatic mutations, including in *KRAS* and *PIK3CA*, in “normal” endometrium of women biopsied for reasons unrelated to endometriosis and adenomyosis. Furthermore, the Gad Getz group recently showed, based on RNA-seq data analyses of many normal tissues (including uterus), that these normal tissues can bear a high burden of somatic mutations (*Science* 2019; doi:10.1126/science.aaw0726). The biological significance of somatic mutations in normal tissues is currently a very hot topic in the field and important for our understanding of human genomic biology and cancer development.

To bolster our hypothesis that an altered/elevated frequency of *KRAS* mutation in NE is a risk

factor/molecular mechanism of adenomyosis pathogenesis, we attempted to collect NE, normal myometrium (NM) and/or endometriosis tissues from not only adenomyosis individuals but also from as many individuals without adenomyosis as possible. In total, we acquired either fresh-frozen or FFPE uterine tissue samples containing NE and NM from 56 individuals (18 from adenomyosis cases, 14 from endometriosis without adenomyosis cases, and 24 from patients (e.g: cervical cancer, ovarian cancer or leiomyoma) with neither adenomyosis nor endometriosis). It should be noted that, although we also collected samples of endometriosis lesions, we were unable to isolate the epithelial component from most FFPE sections, probably because the epithelial component had been lost during surgery. Thus, we were unable to perform genomic analyses of endometriosis lesions. Nevertheless, we were able to enrich for histologically normal epithelial component of endometrium (NE) and normal myometrium (NM) by macro-dissection, and these samples were subjected to TDS to establish their *KRAS*, *PIK3CA* and *PPP2R1A* status (**new Figure 5 and Tables S23–26**).

We divided our study participants into three groups: women with adenomyosis (Group A), women with endometriosis but not adenomyosis (Group E), and women with neither adenomyosis nor endometriosis (Group Non-A/E). *KRAS* mutations were commonly observed in NE of all three groups: Group A (10/18=55.56%), Group E (7/14=50%) and Group Non-A/E (7/24=29.17%) (**new Figure 5A and Table S24**). Similarly, *PIK3CA* mutations were frequently observed in NE of Group A (2/18=11.11%), Group E (5/14=35.71%) and Group Non-A/E (6/24=25%) (**new Figure 5A and Tables S23–24**). These observations are consistent with recent publications that show recurrent *KRAS* and *PIK3CA* mutations in the “normal” NE (doi: <https://doi.org/10.1101/561050> and doi: <https://doi.org/10.1101/505685>). More importantly, the frequencies of *KRAS* driver mutations in group A and group E were significantly higher than in group Non-A/E, supporting our hypothesis that adenomyotic clones originate in NE.

Next, we compared of the VAFs of *KRAS*, *PIK3CA* and *PPP2R1A* driver mutations in Group A or Group E to Group Non-A/E (**new Figure 5B**). The VAFs of mutations encoding oncogenic *KRAS* G12/G13 alterations (*KRAS* p.G12/G13) (but no other mutations) were significantly increased in Group A compared to Group Non-A/E (Mean value of VAF (%): Group A :1.641% vs Group Non-A/E: 0.301%; Welch’s *t*-test:

$p = 0.008$), raising a possibility that *KRAS*-mutated clones had expanded in the NE of individuals with adenomyosis. Taken together, our genomic analyses of NE suggest that an increased frequency of *KRAS*-mutated clones in adjacent “normal” endometrium could be an early step in the molecular pathogenesis of adenomyosis. All these data are presented in our revised Results on Page 15, Lines 5.

Although our study does not precisely characterize the molecular mechanism by which *KRAS*-mutated clones acquire additional “disease-causing” alterations, we feel that such an involved project is beyond the scope of this study, and we look forward to presenting a separate examination of such mechanisms in a future report. In addition, although our study did not definitively identify a common or first event of genetic and/or epigenetic alteration establishing NE clonality, from a mechanistic point of view, *KRAS* or *PIK3CA* mutation would be considered likely to provide NE cell clones with the ability to proliferate and invade neighboring NM and/or other organs, driving the development of adenomyosis. We mention this possibility in our revised Discussion on Page 22 line 1 and Page 22 Line 16, along with the observation that total and cancer-driver mutation burdens in NE are elevated by aging, high body mass index and parity (doi: <https://doi.org/10.1101/505685>). Although we did not observe significant correlations between *KRAS* mutations and age or parity (**Table S25**), it would be intriguing to determine whether these factors are also relevant in the contexts of adenomyosis and endometriosis. Answering this question will require future studies employing a larger patient cohort.

2) Table S6 shows a long list of 134 SNVs detected in adenomyosis tissues, but it does not give an indication of their frequency.

We apologize for not clearly presenting the allele frequency in original Table S6. We did show the mutated allele frequency in column J (“#Alt” for adenomyotic lesions) and column N (“#Alt” for germline control). To enable readers to understand our mutation data more easily, we have changed the relevant heading to “VAF (%)” (instead of “#Alt”) in all revised Tables, including **revised Table S6**.

3) **Without any discussion of these SNVs, the KRAS, PIK3CA, and PPP2R1A mutations are selected in presentation of Fig 1 and the text. Were these the most common?**

We apologize for not describing our work properly and not presenting Figure 1 more clearly. As we noted in the Introduction (Page 5, Line 15 to Page 6, Line 5), recurrent mutations of *KRAS*, *PIK3CA*, *ARID1A*, *PPP2R1A*, *PTEN*, *TP53*, and/or *MED12* occur in uterine endometrial cancer, endometriosis and/or leiomyoma. In our WES study of our discovery cohort (51 individuals; **Figure 1C, pink sequencing column**), recurrent mutations were found only in *KRAS* and in no other genes associated with uterine cancer or disease. TDS analyses of the major uterine disease-associated mutations (*KRAS*, *PIK3CA*, *ARID1A*, *PPP2R1A*, *TP53*, *MED12*) showed that only mutations in *KRAS*, *PIK3CA* and *PPP2R1A* were detected in adenomyosis (**Table S6**). Thus, we presented *KRAS*, *PIK3CA* and *PPP2R1A* as related to gynecologic cancers or diseases in **Figure 1A**. This result was confirmed by our NGS analyses, in which no other recurrent mutation was identified.

We wondered if coverage of our WES was not deep enough to identify all SNVs based on our criteria (mutant read number less than 8) (**Figure S3**) due to either low content or limited expansion of mutated adenomyosis clones in the adenomyosis lesions. To sharpen our analyses, we applied the ClinVar database (<https://www.ncbi.nlm.nih.gov/clinvar/>) to investigate whether any registered pathogenic/driver mutation residues of *KRAS*, *PIK3CA*, *PPP2R1A*, *ARID1A*, *PTEN* and *TP53* could be detected in our WES adenomyosis data, including in the small number of mutant reads that had been filtered out by our SNV criteria (**Figure S3**). We did indeed detect mutation reads of *KRAS*, *PIK3CA*, *PPP2R1A*, *ARID1A*, *TP53* and *PTEN*, with the majority being less than 4 (**Table S10**). To determine if these “candidate mutations” were SNVs or pseudo-positives, we performed TDS on adenomyotic lesions and on as many germline controls as possible. Only *KRAS* and *PIK3CA* alterations were validated as recurrent somatic pathogenic mutations in adenomyosis (**Table S11–12**). When we applied TDS analysis of *KRAS*, *PIK3CA* and *PPP2R1A* to not only our discovery cohort (n=51; **Figure 1C, pink sequencing column**) but also to our additional 19 adenomyosis patients (**Figure 1C, yellow sequencing column**), recurrent somatic pathogenic *KRAS*

mutations, including 25 cases at G12 and one case at Q61, were identified in 37.1% (26/70) of adenomyosis patients (**Figures 1C, 1F–G, Tables S10 and S12**). Somatic mutations encoding the *PIK3CA* H1047 alteration were also validated in lesions from Patients #1 and #27, as was a mutation encoding *PPP2R1A* P179 in a lesion from Patient #2 (**Figures 1C, Tables S10 and S12**). We therefore hypothesized that somatic *KRAS* mutation might be a recurrent driver of adenomyosis. Consistent with published data^{18,20,21}, samples of co-occurring endometriosis in some of our adenomyosis cases bore both *KRAS* and *PIK3CA* mutations (**Figure 1D and Table S12**), whereas most co-occurring leiomyoma samples harbored *MED12* mutations (**Figure 1E**). Thus, the profiles of the dominant mutations driving these three benign gynecological disorders differ. These observations appear in our revised manuscript on Page 9, Line 14.

4) **In this context, Fig 1 is also unclear. The number of SNVs are presented on the y-axis in Fig 1C for each patient ID, with mutation status for the three above genes underneath. However, there are patient IDs with 0 SNVs (no grey squares), which appear to have KRAS mutations. Also, page 8 line 3 states that 134 SNVs were detected in 31/51 adenomyosis cases, but should this not be 31/70 (Fig 1)? This means that the majority of adenomyosis samples actually did not carry a mutation.**

Once again, we apologize for not presenting Figure 1 clearly. We conducted WES analyses on lesions from 51 adenomyosis patients as a discovery cohort (Sequencing; colored **Pink**) and identified somatic mutations in *KRAS*, *PIK3CA* and *PPP2R1A*. We then performed TDS of the *KRAS*, *PIK3CA* and *PPP2R1A* genes in an additional 19 adenomyotic individuals (Sequencing; colored **Yellow**). Thus, in total, 70 adenomyotic patients were examined in this NGS study (Please also see in **Figure S1**). As the reviewer pointed out, there are some patients (for instance, Patients #14–#26) with 0 SNVs (no grey squares) but who harbor *KRAS* mutations. WES analysis did not detect these mutations in these patients most likely because of their low allele frequency and/or insufficient read coverage during WES (coverage: ~135). However, TDS (coverage: ~100,000) was sensitive enough to detect these *KRAS* mutations in these adenomyotic lesions (**Table S12**).

As noted on Page 8, Line 3 of our original manuscript, 134 SNVs were detected in 31/51 adenomyosis cases analyzed by WES. As the reviewer pointed out, around 37%, less than half, of adenomyosis patients carry *KRAS* mutations. This frequency of *KRAS* mutation is comparable to or slightly higher than that in endometriosis (Reference ID: 20 and 21). We describe these observations in our revised manuscript on Page 11, Line 4.

5) **Page 10, lines 7-12: the authors suggest that, because the mutational signatures detected in adenomyosis ‘most closely resemble COMIC signatures associated with ageing’ that this suggested adenomyosis is an age-associated disease. Did the mutational frequency/burden in their dataset associate with age?**

We thank for the reviewer for this constructive question. We tested whether there was a correlation between number of SNVs and age but found none (see Figure below; $R^2=0.0004$). As reviewer #2 criticized our method of signature analysis (major point 8), we accept that our mutational signature analyses data are not meaningful and have removed them from our revised manuscript. **(Please see our response to reviewer #2 major point 8)**. However, we believe that this deletion does not affect the major conclusions of our study.

6) Page 12, lines 11-13. ‘Our data collectively imply that KRAS-mutated clones arising in normal endometrium acquire enhanced invasiveness and proliferative capacity that enables them to grow ectopically, driving both adenomyosis and endometriosis’. **This cannot be concluded, as the mutational frequency of endometrium in non-adenomyosis/non-endometriosis women was not investigated.**

We apologize for our overstatement and agree with the reviewer’s point-of-view. In response to the reviewer’s comment, we conducted TDS of *KRAS* in NE from women with adenomyosis, endometriosis, or neither disease. As described above (major comment 1), the VAFs of *KRAS* mutations in adenomyosis (group A) is significantly higher than in women without adenomyosis/endometriosis (group Non-A/E) (**new Figure 5B**), suggesting that *KRAS*-mutated clones have expanded in patients with adenomyosis. In our revised manuscript, we state on **Page 15, Line 1** that: “Our data collectively raise the possibility that *KRAS*-mutated clones arising in NE acquire enhanced invasiveness and proliferative capacity that enable them to grow ectopically, driving adenomyosis.”. We also refer the reviewer back to our response to his/her major comment #1, in which we cite our data showing that clones with *KRAS* mutations seem to have highly expanded in NE adjacent to adenomyotic lesion but not in women without adenomyosis or endometriosis.

Minor comments

Page 4 line 9 states that differences exist in pathogenesis between adenomyosis and endometriosis, whereas page 5 line 1-3 states that a common molecular mechanism is hypothesised. This reads as contrasting information.

We apologize for not clearly describing the differences between the pathogenic and molecular mechanisms of adenomyosis and endometriosis. We intended to state that adenomyosis and endometriosis “SHARE”

some histological findings and molecular changes, but that there are also some “DIFFERENCES” in their pathogenesis, lesion localization and clinical features (Page 4, line 9 in the previous manuscript). In our revised manuscript on **Page 5 Line 3**, we clarify that the recent review by Koninckx *et al.* HYPOTHESIZED that a common mechanism might underlie the pathogenesis of endometriosis and adenomyosis but did so WITHOUT any direct evidence. Thus, whether a molecular mechanism is shared by adenomyosis and endometriosis has been UNKNOWN due to a lack of genomic analyses of adenomyosis.

Page 6, line 1. When referring to the published mutational results for endometriosis, it is important to add a word of caution that the extent to which these are related to disease mechanism is unknown (see earlier comment re. high frequency of somatic mutation burden in ‘normal’ tissues, including endometrium)

Once again, we sincerely appreciate the reviewer’s point and have now mentioned the two online preprint repository reports. We also recognized the related publication by Dr. Gad Getz’s group in *Science*. In our revised manuscript, we state on **Page 6 Line 7** that: “However, very recent genomic analyses of normal human tissues, including endometrium, have undermined this hypothesis. RNA sequencing analyses of 29 types of normal tissues, including uterus, have uncovered the presence of expanded clones of cells with relatively high mutation burdens in many normal tissues²². Furthermore, endometrium samples from individuals who have undergone endometrial biopsy for a reason unrelated to endometriosis have frequently shown *KRAS* and *PIK3CA* mutations^{23,24}. Thus, endometrial clones bearing *KRAS* and/or *PIK3CA* mutations may not necessarily drive the pathogenesis of endometriosis. Ideally, the status of *KRAS*- and/or *PIK3CA*-mutated clones in histologically normal endometrium should be compared between individuals with and without endometriosis. Accordingly, the exact contribution of *KRAS* and *PIK3CA* mutations to endometriosis remains unresolved.”

Page 7, line 4-6: a reference should be made to the methods section for description of clinical

sampling. It should be made clear if all patients underwent MRI for diagnostic purposes, and also how ‘adjacent normal’ tissues were confirmed to be normal: through histology? How was ‘adjacent’ defined/chosen?

We apologize for not clearly describing how we acquired our clinical samples for this study. As the reviewer assumed (and is now stated in our revised manuscript on Page 25 Line 11, “All patients underwent MRI for diagnostic purposes. Adjacent normal tissues, which were grossly and tactically distinguishable from adenomyotic lesions, were collected during surgery. Histological review by the study pathologist confirmed that there was no significant contamination of normal tissue by adenomyotic cells. See also Tables S1–2 and S23–24 for additional clinical details pertaining to these specimens.”.

Page 9, line 3: Fig 1G is referenced but does not exist.

We apologize for providing incorrect information in our original manuscript. Figure 1G, its legend and its call-out in the main text on Page 11 Line 8 are now properly included in our revised manuscript.

Page 13, line 3: ‘KRAS mutations were more frequent in patients who have been pre-treated with the PR agonist DNG... raising the possibility that DNG is less efficacious in KRAS-mutated disease’. How this conclusion is made is currently unclear. Presumably, the assumption is that women undergoing surgery for adenomyosis in the present study were doing so, because their treatment was ineffective; i.e. the surgery was an alternative to medication. This should be made clear in the methods. Currently, ‘pre-treatment with DNG’ to the uninitiated reader could mean a standard treatment prior to surgery, which then begs the question why the above conclusion is made.

We apologize for not describing this issue clearly. As the reviewer supposed, the patients were initially treated with DNG. The patients who did not respond to DNG underwent surgery as an alternative therapy. In

our revised manuscript, we state on **Page 17, Line 8**: “Many adenomyotic patients initially receive hormonal treatment, including administration of DNG. Those patients who do not respond to such hormonal treatment then undergo surgery. Our analyses revealed intriguing clinical differences between cases of adenomyosis with *KRAS* mutations versus those without. Specifically, *KRAS* mutations were more frequent in lesions of patients who had been pretreated with the PR agonist DNG^{17,25} ($p = 0.0002$) but not with GnRH α (Tables 1 and S22). This observation raises a possibility that DNG may be less efficacious in patients bearing *KRAS*-mutated adenomyotic clones, allowing these abnormal cells to persist until surgery.”.

Response to Reviewer #2

We thank for the reviewer for his/her positive comment: “**The molecular analyses of a large series of uterine adenomyosis specimens is valuable and important.**”. We appreciate the insights and constructive suggestions provided by the reviewer.

Reviewer #2’s Specific Comments

1) **Page 7, line 16 states “Our robust criteria and validation by targeted deep sequencing (TDS) (see Materials and Methods, 1 Figure S3) permitted us to detect 134 unique synonymous and non-synonymous single nucleotide variations (SNVs) in 31/51 (60.8%) adenomyosis cases, defining adenomyosis as a clonal disorder with somatic mutations (Tables S5–6).” Reviewer comment: The conclusion that adenomyosis is a “clonal” disorder is not supported by the aforementioned observation.**

We apologize for our overstatement. We accept that our NGS study alone is not sufficient for defining that adenomyosis is a clonal disorder. However, we believe that our combination of NGS plus our LCM experiments (**Figure 2**) show that adenomyosis is a clonal disorder with somatic mutations. In our revised manuscript, we state on **Page 9, Line 2** that: “Our robust criteria and validation by targeted deep sequencing

(TDS) (see Materials and Methods, Figure S3) permitted us to detect 134 unique synonymous and non-synonymous single nucleotide variations (SNVs) in 31/51 (60.8%) adenomyosis cases, raising the possibility that adenomyosis is a clonal disorder with somatic mutations (Tables S5–6).”

2) Page 8, line 6: A mean VAF of 4.8% is noted. The accompanying range should be provided.

We thank the reviewer for this suggestion. In our revised manuscript, we note that “These adenomyosis SNVs were present in low numbers (mean of 2.6 mutations/individual) and at low variant allele frequencies (VAF) (mean 4.8%; range 2.47–16.02%).” (Page 9, Line 8).

3) Page 8, line 14 states: “We next used the ClinVar database (<https://www.ncbi.nlm.nih.gov/clinvar/>) to select our mutation candidates for further study (see Materials and Methods, Table S10).”

Reviewer comment: There is no information describing this methodology in either the “Materials and Methods” or in the “Supplemental Materials and Methods”. This information needs to be provided in order to review the validity of the strategy used to choose candidate mutations from all SNVs and to understand why the authors do not mention or focus in the recurrent PIK3CA, TP53, and ARID1A mutations listed in Table S12. Also, the authors need to define the meaning of “mutation candidate”; do they mean candidate driver mutation?

We apologize for not clearly describing the methodology of the mutational analyses used in our study. We first ascertained if any uterine endometrial carcinoma and endometriosis-associated genes (*KRAS*, *PPP2R1A*, *PIK3CA*, *ARID1A*, *PTEN* and *TP53*) (Reference ID: 18–21) were mutated in adenomyosis. Our WES analyses detected *KRAS* mutations in samples from three individuals and a *PPP2R1A* mutation in a sample from one patient, but no *PIK3CA*, *PPP2R1A*, *ARID1A* or *TP53* mutations were called as SNVs according to our criteria (Table S6). We wondered if coverage of our WES was not deep enough to identify all SNVs based on these criteria (mutant read number less than 8) (Figure S3) due to either low content or

limited expansion of mutated adenomyosis clones in the adenomyosis lesions. To sharpen our analyses, we applied the ClinVar database (<https://www.ncbi.nlm.nih.gov/clinvar/>) to investigate whether any registered pathogenic/driver mutation residues of *KRAS*, *PIK3CA*, *PPP2R1A*, *ARID1A*, *PTEN* and *TP53* could be detected in our WES adenomyosis data, including in the small number of mutant reads that had been filtered out by our SNV criteria (**Figure S3**). We did indeed detect mutation reads of *KRAS*, *PIK3CA*, *PPP2R1A*, *ARID1A*, *TP53* and *PTEN*, with the majority being less than 4 (**Table S10**). To determine if these candidate mutations were SNVs or pseudo-positives, we performed TDS on adenomyotic lesions and on as many germline controls as possible. Only *KRAS* and *PIK3CA* alterations were validated as recurrent somatic pathogenic mutations in adenomyosis (**Table S11–S12**). When we applied TDS analysis of *KRAS*, *PIK3CA* and *PPP2R1A* to not only our discovery cohort (n=51; **Figure 1C, yellow sequencing column**) but also to our additional 19 adenomyosis patients (**Figure 1C, pink sequencing column**), recurrent somatic pathogenic *KRAS* mutations, including 25 cases encoding at G12 and one case encoding at Q61, were identified in 37.1% (26/70) of adenomyosis patients (**Figures 1C, 1F–G, Table S12**). Somatic mutations encoding the *PIK3CA* H1047 alteration were also validated in lesions from Patients #1 and #27, as was a mutation encoding *PPP2R1A* P179 in a lesion from Patient #2 (**Figures 1C, Table S12**). We therefore hypothesized that somatic *KRAS* mutation might be a recurrent driver of adenomyosis. Consistent with published data^{18,20,21}, samples of co-occurring endometriosis in some of our adenomyosis cases bore both *KRAS* and *PIK3CA* mutations (**Figure 1D and Table S12**), whereas most co-occurring leiomyoma samples harbored *MED12* mutations (**Figure 1E**). Thus, the profiles of the dominant mutations driving these three benign gynecological disorders differ. These observations appear in our revised manuscript on **Page 9, Line 14**. Once again, we apologize for our lack of our clarity.

4) Page 9, line 2 states “ As non-KRAS pathogenic mutations were not linked to adenomyosis using this strategy....” Reviewer comment: What “strategy” are the authors referring to, and what is the intended meaning of “linked”?

We apologize for not describing our meaning properly. Please see our response to major comment 3). In our revised manuscript, we have deleted the original sentence beginning “As non-KRAS pathogenic mutations were not linked to adenomyosis using this strategy....”.

5) Table S12: The Tab for this table in the Excel spreadsheet is “Table S12 PTENARID1Adeepseq”, but there is no information on PTEN in the table (the table shoes data for PIK3CA, TP53, and ARID1A).

We apologize for this careless mistake. The *PTEN* data now appear in revised Table S11.

6) The rationale for, and interpretation of, the experiments in this section are disconnected. The rationale states “Only a minor fraction of an adenomyosis lesion is comprised of epithelial cells (Figure 2A). This observation prompted us to consider whether the low VAFs we observed were due to low epithelial component content or limited clonal expansion of mutated cells.” After sequencing DNA from laser capture microdissected epithelial and myometrial cells, the study findings and conclusions are “Compared to bulk frozen samples, VAFs were markedly increased in all three LCM-A samples (Figures 2C–E, S6, Table S13), demonstrating that somatic mutation occurs in the epithelial component of adenomyosis. Thus, adenomyosis may arise from the ectopic proliferation of epithelial cell clones”. **Comment: The authors need to provide an interpretation of their findings in relation to their initial question of whether the low VAFs in unmicrodissected samples were due to low epithelial component content or limited clonal expansion of mutated cells. They also need to state and discuss on the fact that they detected mutations, including KRAS and PPP2R1A mutations, in the laser captured adjacent normal muscle cells (Figure 2C-E).**

We thank the reviewer for these valuable suggestions to improve our manuscript. We believe that our LCM experiments adequately demonstrate that the low VAFs in un-microdissected samples (**Bulk-A shown in**

Figure 2C–E) were due to low epithelial component content rather than limited clonal expansion of mutated cells. We also respectfully disagree with the reviewer’s comment; **“They also need to state and discuss on the fact that they detected mutations, including KRAS and PPP2R1A mutations, in the laser captured adjacent normal muscle cells (Figure 2C-E)”**. In Figure 2C–E, all mutations, including *KRAS* and *PPP2R1A* mutations, were markedly increased in the laser captured epithelial component of adenomyosis (LCM-A) but NOT in laser captured adjacent normal muscle cells (LCM-ADJ). We accept and apologize that we did not make this difference clear in our original text. In our revised manuscript, we state that: “We isolated genomic DNA from epithelial cells of adenomyosis tissue (LCM-A), as well as from adjacent muscle cells (non-diseased control; LCM-ADJ), and performed TDS. Compared to bulk frozen samples, VAFs were markedly increased in LCM-A (but not LCM-ADJ) samples from all three adenomyosis cases (Figures 2C–E, S6, Table S13), demonstrating that the low VAFs detected in our WES analyses of bulk frozen adenomyosis lesions were due to low epithelial component content rather than to poor expansion of mutated adenomyosis clones. These LCM experiments indicate that somatic mutation occurs in the epithelial component of adenomyosis, and that adenomyosis may thus arise from the ectopic proliferation of the mutated epithelial cell clones.” (Page 12, Line 9).

7) **It is unclear how meaningful the analysis of mutational signatures is since it is based on an analysis only 148 variants and all variants were pooled and analyzed as a single set (rather than separately assessing the mutational signature of variants in each lesion). In addition, the finding that the pooled mutational signature most closely resemble COSMIC aging signatures is puzzling because the patients in this study are relatively young (range 21yr-58yr; mean 42.5yr for KRAS-wildtype cases; mean 41yr for KRAS-mutated cases) (Table 1). The authors do discuss the latter point.**

We thank the reviewer for this criticism of our mutational signature analyses. As the reviewer pointed out, such analyses ideally should be performed by separately assessing each sample and not by pooling all variants into a single set. However, as we described in our original Supplementary Materials and Methods,

we HAD to pool all SNVs from 31 adenomyosis patients and consider them as being in a single individual because the number of mutations in adenomyosis is too low to perform the mutational signature analysis. To avoid misleading the reader, we have removed these data from our revised manuscript, but strongly believe that the major conclusions of our study are not affected by a lack of mutational signature analyses.

8) The authors need to indicate what the variable blue coloring symbolizes in the Tables shown in in Figure 3 and 4.

We thank the reviewer for this helpful advice. As we describe in the **revised legend of Figures 3 and 4**, the shade of the blue coloring symbolizes the relative VAF value of each sample compared to the maximum VAF value. The darkest blue represents the maximum VAF value within the total sample and the white column represents the absence of a mutation in that gene. For instance, **Figure 4A** shows a TDS evaluation of 6 multi-sampled adenomyosis lesions (1-6), adjacent normal endometrium (NE), adjacent normal myometrium (NM), co-occurring leiomyoma (LM) and peripheral blood (B). The highest VAF of a mutation in *ARHGAP42* was detected in adenomyosis sample 1 (see **Table S17**; 3.884% VAF in the sample 8A1), which represents 100% and is shown in darkest blue. Mutations in *ARHGAP* with relatively high VAF values were detected in adenomyosis samples 3 and 5 (see **Table S17**; 1.571 and 1.601% in the sample 8A3 and 8A5, respectively), and are shown in a slightly lighter shade of bright blue in these columns. Marginal VAF values for *ARHGAP* mutations were detected in the remaining adenomyosis samples as well as in non-adenomyosis samples (NE, NM, LM and B), which therefore are shown in white in these columns. In our revised manuscript, in the **legend of Figure 3 and 4** on **Pages 49 and 51**, respectively, we have replaced “Ratio (%)” with “Relative VAF(%)” and state that: “VAF values are shown using a color scale where dark blue is the maximum VAF value within the total sample and white is the absence of a mutation in that gene. The VAF of a mutation in a sample relative to the highest VAF value is represented as a shade of a bright blue color.”

9) Page 11, line 1 states “Some mutations were shared among almost all multi-regional samples from a single patient [e.g : ZNF672 in #28 (Figure 4 3A); C1QTNF and MSS51 in #29 (Figure 3B)], suggesting that these genetic alterations were acquired early during adenomyosis development, while other mutations were more restricted (Figure 3). Most mutations detected in adenomyosis and co-existing leiomyoma were mutually exclusive (Figures 3B–C, 4A, Tables S15–17), implying the lack of a clonal relationship between these disorders. Thus, despite their frequent co-occurrence, adenomyosis and leiomyoma are distinct entities.” Comment: The statistical analyses provided for variants in Tables S15, S16 and S17 (column O) seems to indicate that most mutations are not considered somatic. Therefore, it is unclear if the authors are assigning a different interpretation to variants when they are displayed in Tables S15, S16 and S17 versus when they are displayed in the boxes in Figure 3 and Figure 4. These points require clarification in the manuscript.

We apologize for not presenting our statistical analyses clearly. These analyses apply to not only Tables S15–17 but also Tables S12, 14, 18–20 and 28–29. However, we respectfully disagree with the reviewer most mutations are not considered somatic. In our revised Materials and Methods section, we state that: “As the VAF values of the observed mutations in adenomyosis were quite low, we performed a statistical calculation to determine whether these alterations were true somatic mutations or NGS noise. In general, it is expected that the distribution of the difference in VAFs between lesion and control samples follows a normal distribution. For each position i , a difference of VAFs between adjacent (a) and normal (n) is defined by $d_i = \max_{y \in \Sigma} |a_{y,i} - n_{y,i}|$, where Σ is the set of nucleotides $\{A, C, G, T\}$. Let μ_j and σ_j be the mean (shown in column M in Table S12 and column L in Tables S14–20 and S28–29) and the standard deviation (shown in column N in Table S12 and column M in Tables S14–20 and S28–29), respectively, of the set $\{d_k \mid k = j - K, \dots, j - 1, j + 1, \dots, j + K\}$ for the estimated position j . If one assumes that $X \sim N(\mu_j, \sigma_j^2)$, then the somatic mutation rate for j is defined by $P(X > d_j) < 0.05$, $d_j > 0.001$ and $n_{y,j} < 0.01$, where $y \in \Sigma$. For K values, we used the values shown in length (bp) column O in Table S12

and column N in Tables S14–20 and S28–29 for the individual mutation analyses. For assessment of statistical significance of somatic mutations, the *p*-value of each mutation is shown in column P in Table S12 and column O in Tables S14–20 and S28–29.” (Page 29, Line 5).

10) The manuscript states “KRAS mutations were more frequent in patients who had been pretreated with the PR agonist DNG17,22 ($p = 0.0002$, Table 1), raising the possibility that DNG is less efficacious in KRAS-mutated disease.” The latter comment is not warranted since the authors do not provide any data on treatment outcome (efficacy) of DNG treatment for their adenomyosis patients. They also do not clarify whether these patients received DNG treatment for their adenomyosis or for co-existing endometriosis. If the treatment was based on co-existent endometriosis this may actually be the feature that is driving the enrichment for KRAS mutations in DNG-treated adenomyosis cases, since this study reports shared KRAS mutations between co-existing adenomyosis and endometriosis.

We thank the reviewer for these thoughtful comments and solid suggestions. We agree that our statement that DNG may be less efficacious in *KRAS*-mutated disease is based on our *in vitro* studies of immortalized cells (original Figure 5 and new Figure 6) and has yet to be validated by analyzing the clinical outcomes of the DNG-treated adenomyosis patients in this study. Efficacy of DNG as a treatment for adenomyosis has been previously reported (references 34 and 35). However, in our study, the clinical assessments of DNG-treated adenomyosis patients were carried out at two independent hospitals with differing assessment protocols and equipment. We felt that combining clinical data from these two different hospitals with differing approaches might be misleading. We totally concur with the reviewer that such correlations properly done are necessary for the translation of our study’s findings to clinical application. However, we feel such detailed validation is beyond the scope of our present genomic study. A future validation study that is based on appropriate data collection from a reasonably sized cohort should be performed. In our revised Discussion, we state that: “Our work using an immortalized cell system *in vitro* has suggested that adenomyotic lesions frequently contain *KRAS* mutations that may epigenetically downregulate *PR* and thus

reduce DNG efficacy. This observation should be followed up in a large number of DNG-treated adenomyosis patients to determine if our hypothesis is valid and if its implications are translatable to the clinic.” (Page 24, Line 1).

In answer to the reviewer’s second point, DNG was administered to adenomyosis patients without (3 of 22; 13.6%) and with (10 of 48; 20.8%) co-occurring endometriosis. A Fisher exact *t*-test analysis confirmed that there was no significant difference between these two groups (*p*-value: 0.7415). Thus, we do not think that DNG treatment drives any enrichment of *KRAS* mutations in DNG-treated adenomyosis cases.

11) Table 1 shows the frequency of DNG treatment for *KRAS* wildtype versus *KRAS* mutated cases. The reader has to use these data to manually calculate the frequency of *KRAS* mutations in patients pretreated with DNG (11 of 13, 84%) and in untreated patients (15 of 57, 26%).

We apologize for not presenting our Table 1 data more conveniently. As 26 different patient characteristics were analyzed in Table 1, we opted to present only DNG-treated (and not untreated) patient numbers in order to summarize the data. In our revised manuscript, we include not only positive (DNG-treated, PR-negative or EN co-occurrence) but also negative (DNG-untreated, PR-positive, or adenomyosis alone) clinical characteristics that showed statistically significant differences. These data now appear in **new Table S22**.

12) The concentration of DNG used in the cell-based experiments is not provided. The vehicle (DMSO? Other?) in which DNG was resuspended is not provided. For the cell-based assays the comparison seems to be DNG-treated cells versus untreated cells. It should be DNG-treated versus vehicle-treated cells. Until the data on DNG- and vehicle-treatments data are provided the validity of the cell-based assays cannot be determined.

We thank the reviewer for this suggestion and apologize for not clarifying that our control was indeed

vehicle (DMSO) treatment. In our revised manuscript, we include this information in **the legend of new Figure 6** on **Page 54** and also note that, for panel B, results were expressed as the percentage of viable cells relative to vehicle control cells. Similarly, for panel C, results were the percentage of BrdU⁺ cells relative to vehicle control cells.

13) The Life Sciences study design information states “Data reproducibility was examined by 3 independent experiments in cell viability (Figure 5A).” Were technical replicates included within each of the three experiments?

We thank the reviewer for this opportunity to improve our data reproducibility. For original Figure 5A, we counted cell numbers in triplicate for each of three independent experiments. However, we recognize that this number of independent experiments is too low to provide convincing reproducibility in the cell line data at the original manuscript. After the submission of our original manuscript, we performed additional independent experiments and now confirm that our data are reproducible. We present these additional data in a new **Figure 6** on **Page 54** and clarify the number of experiments and replicates in its legend.

14) The data displayed in Figure S16 should be included in the main Figure 5 since a major claim of the paper is that mutant KRAS downregulates PR expression. Having said that, I do not find the observed downregulation of PR expression by KRAS-mut (Figure 16B) very convincing. What exactly are the P values for these data? The fold difference in expression seems to shift from ~1.0 to ~0.5.

We thank the reviewer for this helpful suggestion. In our revised manuscript, we now present the data of original Figures 5 and S16 as a **revised Figure 6**. We apologize for not presenting our qPCR results clearly. The data are levels of *PR-A/B* mRNA expression in cells expressing *KRAS-WT*, *KRAS-Mut*, *PIK3CA-WT* or *PIK3CA-Mut* relative to that in cells transfected with empty vector (control samples). Thus, the mean values and standard deviations represent the fold change in *PR-A/B* mRNA level compared to that in controls

(empty vector transfected samples). The statistical significance (p -values) of differences in these values was assessed by the Welch's t -test. To increase our data reproducibility, we have repeated our qPCR analyses. In our revised manuscript, we present qPCR analyses of triplicate samples in each of six independent experiments. We have confirmed that there is statistically significant downregulation of *PR-A/B* mRNA in *KRAS*-Mut (G12C or G12D)-expressing samples compared to empty vector-transfected or *KRAS*-WT-expressing samples as assessed by the Welch's t -test (p -values <0.05). The raw data underlying these statistical analyses are included in the Source Data file. In our revised manuscript, we have refined the layout of new Figure 6D and state in its legend: "(D) Quantitative RT-PCR determination of mRNA levels of *PR-A/B* in the immortalized cells in (A). Values were normalized to *GAPDH*. Data are the mean fold change \pm SD relative to levels in control cells stably infected with empty vector. Six independent experiments, each with three technical replicates per group. $*p < 0.05$ by Welch's t -test. Source data are provided as a Source Data file." (Page 55).

15) There are numerous errors and inconsistencies in the descriptions of Table S1-S12. The authors need to carefully check all tables descriptions for accuracy. As examples:

i) Tab for Table S3 is "Table S3 clinical info", whereas the header for Table S3 is "Table S3. Method used to detect genomic alterations in tissue samples from all patients."

ii) Tab for Table S12 is "Table S12 PTENARID1Adeepseq", whereas the header for Table S12 is "Table S12. Summary of TDS read information for pathogenic mutations defined by the ClinVar database."

We apologize for our careless mistakes in listing our Supplementary Tables. We corrected these errors in our revised manuscript.

Reviewers' comments:

Reviewer #1 (Remarks to the Author):

The authors have significantly enhanced their paper, by adding new data and addressing many of the comments made by both reviewers. Specifically, they have responded to the criticism that they had not included histologically normal endometrium (NE) and myometrium (NM) as a comparison for both adenomyosis cases and controls. Their new analyses, presented in a new figure are a very good addition to the paper. They show an apparent significant increase in KRAS mutational load comparing NE from adenomyosis cases vs. controls, although this was not the case for PIK3CA or PPP2R1A. This should be made clearer on page 16, lines 10-13. 'Importantly, KRAS mutations were more often detected in group A and group E than in group Non-A/E (P=xxxx), BUT NOT IN PIK3CA OR PPP2R1A, supporting our hypothesis that an increased frequency of KRAS-mutated clones in NE might be the origin of adenomyosis'.

Page 5, line 6: 'However, no direct evidence to support this hypothesis exists at present due to a lack of genomic analyses of adenomyosis'. It is not just genomic analysis that could elucidate common molecular mechanisms of pathogenesis. This should be rephrased.

Throughout the manuscript: there is frequent use of the phrase 'driver mutations' for adenomyosis or endometriosis. I have a problem with the extensive and definitive use of 'driver', as it seems to imply that the mutations are a crucial part of pathogenesis. Yet, they could equally be a consequence/by-product of disease, e.g. as a result of aberrant endometrial proliferation. The author correctly edited this on page 15, line 1 where they state more carefully that the data 'raise the possibility' that KRAS-mutated clones arising in NE acquire invasiveness and proliferative capacity that enable them to grow ectopically, driving adenomyosis. Other instances of reference to 'driver' mutations should be reworded.

The additional text provided is long in places, and would benefit from editorial attention.

Although the manuscript does not answer the question to what extent the somatic mutations observed in KRAS are part of the pathogenic process, the results add significantly to the field and are highly novel for adenomyosis.

Reviewer #2 (Remarks to the Author):

Reviewer-2 comments on the rebuttal are included below. While the authors have responded satisfactorily to most comments, but there are still a few outstanding points, and new points, that need to be addressed.

Response to Reviewer #2

We thank for the reviewer for his/her positive comment: "The molecular analyses of a large series of uterine adenomyosis specimens is valuable and important.". We appreciate the insights and constructive suggestions provided by the reviewer.

Reviewer #2's Specific Comments

1) Page 7, line 16 states "Our robust criteria and validation by targeted deep sequencing (TDS) (see Materials and Methods, 1 Figure S3) permitted us to detect 134 unique synonymous and non-synonymous single nucleotide variations (SNVs) in 31/51 (60.8%) adenomyosis cases, defining adenomyosis as a clonal disorder with somatic mutations (Tables S5–6)." Reviewer comment: The conclusion that adenomyosis is a "clonal" disorder is not supported by the aforementioned

observation.

We apologize for our overstatement. We accept that our NGS study alone is not sufficient for defining that adenomyosis is a clonal disorder. However, we believe that our combination of NGS plus our LCM experiments (Figure 2) show that adenomyosis is a clonal disorder with somatic mutations. In our revised manuscript, we state on Page 9, Line 2 that: "Our robust criteria and validation by targeted deep sequencing (TDS) (see Materials and Methods, Figure S3) permitted us to detect 134 unique synonymous and non-synonymous single nucleotide variations (SNVs) in 31/51 (60.8%) adenomyosis cases, raising the possibility that adenomyosis is a clonal disorder with somatic mutations (Tables S5–6)."

Reviewer-2 comment on rebuttal: Revision is okay, but please check the call-out to Tables S5-6 in the sentence above; I believe this should be Tables S6-7. Please check the accuracy of all other display item call-outs in the text.

2) Page 8, line 6: A mean VAF of 4.8% is noted. The accompanying range should be provided.

We thank the reviewer for this suggestion. In our revised manuscript, we note that "These adenomyosis SNVs were present in low numbers (mean of 2.6 mutations/individual) and at low variant allele frequencies (VAF) (mean 4.8%; range 2.47–16.02%)." (Page 9, Line 8).

Reviewer-2 comment on rebuttal: Revision is okay.

3) Page 8, line 14 states: "We next used the ClinVar database (<https://www.ncbi.nlm.nih.gov/clinvar/>) to select our mutation candidates for further study (see Materials and Methods, Table S10)." Reviewer comment: There is no information describing this methodology in either the "Materials and Methods" or in the "Supplemental Materials and Methods". This information needs to be provided in order to review the validity of the strategy used to choose candidate mutations from all SNVs and to understand why the authors do not mention or focus in the recurrent PIK3CA, TP53, and ARID1A mutations listed in Table S12. Also, the authors need to define the meaning of "mutation candidate"; do they mean candidate driver mutation?

We apologize for not clearly describing the methodology of the mutational analyses used in our study. We first ascertained if any uterine endometrial carcinoma and endometriosis-associated genes (KRAS, PPP2R1A, PIK3CA, ARID1A, PTEN and TP53) (Reference ID: 18–21) were mutated in adenomyosis. Our WES analyses detected KRAS mutations in samples from three individuals and a PPP2R1A mutation in a sample from one patient, but no PIK3CA, PPP2R1A, ARID1A or TP53 mutations were called as SNVs according to our criteria (Table S6). We wondered if coverage of our WES was not deep enough to identify all SNVs based on these criteria (mutant read number less than 8) (Figure S3) due to either low content or limited expansion of mutated adenomyosis clones in the adenomyosis lesions. To sharpen our analyses, we applied the ClinVar database (<https://www.ncbi.nlm.nih.gov/clinvar/>) to investigate whether any registered pathogenic/driver mutation residues of KRAS, PIK3CA, PPP2R1A, ARID1A, PTEN and TP53 could be detected in our WES adenomyosis data, including in the small number of mutant reads that had been filtered out by our SNV criteria (Figure S3). We did indeed detect mutation reads of KRAS, PIK3CA, PPP2R1A, ARID1A, TP53 and PTEN, with the majority being less than 4 (Table S10). To determine if these candidate mutations were SNVs or pseudo-positives, we performed TDS on adenomyotic lesions and on as many germline controls as possible. Only KRAS and PIK3CA alterations were validated as recurrent somatic pathogenic mutations in adenomyosis (Table S11–S12). When we applied TDS analysis of KRAS, PIK3CA and PPP2R1A to not only our discovery cohort (n=51; Figure 1C, yellow sequencing column) but also to our additional 19 adenomyosis patients (Figure 1C, pink sequencing column), recurrent somatic pathogenic KRAS mutations, including 25 cases encoding at G12 and one case encoding at Q61, were identified in 37.1% (26/70) of adenomyosis patients

(Figures 1C, 1F–G, Table S12). Somatic mutations encoding the PIK3CA H1047 alteration were also validated in lesions from Patients #1 and #27, as was a mutation encoding PPP2R1A P179 in a lesion from Patient #2 (Figures 1C, Table S12). We therefore hypothesized that somatic KRAS mutation might be a recurrent driver of adenomyosis. Consistent with published data^{18,20,21}, samples of co-occurring endometriosis in some of our adenomyosis cases bore both KRAS and PIK3CA mutations (Figure 1D and Table S12), whereas most co-occurring leiomyoma samples harbored MED12 mutations (Figure 1E). Thus, the profiles of the dominant mutations driving these three benign gynecological disorders differ. These observations appear in our revised manuscript on Page 9, Line 14. Once again, we apologize for our lack of our clarity.

Reviewer-2 comment on rebuttal: Revision is okay.

4) Page 9, line 2 states “ As non-KRAS pathogenic mutations were not linked to adenomyosis using this strategy...” Reviewer comment: What “strategy” are the authors referring to, and what is the intended meaning of “linked”?

We apologize for not describing our meaning properly. Please see our response to major comment 3). In our revised manuscript, we have deleted the original sentence beginning “As non-KRAS pathogenic mutations were not linked to adenomyosis using this strategy...”.

Reviewer-2 comment on rebuttal: Revision is okay.

5) Table S12: The Tab for this table in the Excel spreadsheet is “Table S12 PTENARID1Adeepseq”, but there is no information on PTEN in the table (the table shoes data for PIK3CA, TP53, and ARID1A).

We apologize for this careless mistake. The PTEN data now appear in revised Table S11.

Reviewer-2 comment on rebuttal: Revision is okay.

6) The rationale for, and interpretation of, the experiments in this section are disconnected. The rationale states “Only a minor fraction of an adenomyosis lesion is comprised of epithelial cells (Figure 2A). This observation prompted us to consider whether the low VAFs we observed were due to low epithelial component content or limited clonal expansion of mutated cells.” After sequencing DNA from laser capture microdissected epithelial and myometrial cells, the study findings and conclusions are “Compared to bulk frozen samples, VAFs were markedly increased in all three LCM-A samples (Figures 2C–E, S6, Table S13), demonstrating that somatic mutation occurs in the epithelial component of adenomyosis. Thus, adenomyosis may arise from the ectopic proliferation of epithelial cell clones”. Comment: The authors need to provide an interpretation of their findings in relation to their initial question of whether the low VAFs in unmicrodissected samples were due to low epithelial component content or limited clonal expansion of mutated cells. They also need to state and discuss on the fact that they detected mutations, including KRAS and PPP2R1A mutations, in the laser captured adjacent normal muscle cells (Figure 2C-E).

We thank the reviewer for these valuable suggestions to improve our manuscript. We believe that our LCM experiments adequately demonstrate that the low VAFs in un-microdissected samples (Bulk-A shown in Figure 2C–E) were due to low epithelial component content rather than limited clonal expansion of mutated cells. We also respectfully disagree with the reviewer’s comment; “They also need to state and discuss on the fact that they detected mutations, including KRAS and PPP2R1A mutations, in the laser captured adjacent normal muscle cells (Figure 2C-E)”. In Figure 2C–E, all mutations, including KRAS and PPP2R1A mutations, were markedly increased in the laser captured epithelial component of adenomyosis (LCM-A) but NOT in laser captured adjacent normal muscle cells (LCM-ADJ). We accept and apologize that we did not make this difference clear in our original text. In our revised manuscript, we state that: “We isolated genomic DNA from epithelial cells of adenomyosis tissue (LCM-A), as well as from adjacent muscle cells (non-diseased control;

LCM-ADJ), and performed TDS. Compared to bulk frozen samples, VAFs were markedly increased in LCM-A (but not LCM-ADJ) samples from all three adenomyosis cases (Figures 2C–E, S6, Table S13), demonstrating that the low VAFs detected in our WES analyses of bulk frozen adenomyosis lesions were due to low epithelial component content rather than to poor expansion of mutated adenomyosis clones. These LCM experiments indicate that somatic mutation occurs in the epithelial component of adenomyosis, and that adenomyosis may thus arise from the ectopic proliferation of the mutated epithelial cell clones.” (Page 12, Line 9).

Reviewer-2 comment on rebuttal: Revision is acceptable.

7) It is unclear how meaningful the analysis of mutational signatures is since it is based on an analysis only 148 variants and all variants were pooled and analyzed as a single set (rather than separately assessing the mutational signature of variants in each lesion). In addition, the finding that the pooled mutational signature most closely resemble COSMIC aging signatures is puzzling because the patients in this study are relatively young (range 21yr-58yr; mean 42.5yr for KRAS-wildtype cases; mean 41yr for KRAS-mutated cases) (Table 1). The authors do discuss the latter point.

We thank the reviewer for this criticism of our mutational signature analyses. As the reviewer pointed out, such analyses ideally should be performed by separately assessing each sample and not by pooling all variants into a single set. However, as we described in our original Supplementary Materials and Methods, we HAD to pool all SNVs from 31 adenomyosis patients and consider them as being in a single individual because the number of mutations in adenomyosis is too low to perform the mutational signature analysis. To avoid misleading the reader, we have removed these data from our revised manuscript, but strongly believe that the major conclusions of our study are not affected by a lack of mutational signature analyses.

Reviewer-2 comment on rebuttal: Revision is okay

8) The authors need to indicate what the variable blue coloring symbolizes in the Tables shown in in Figure 3 and 4.

We thank the reviewer for this helpful advice. As we describe in the revised legend of Figures 3 and 4, the shade of the blue coloring symbolizes the relative VAF value of each sample compared to the maximum VAF value. The darkest blue represents the maximum VAF value within the total sample and the white column represents the absence of a mutation in that gene. For instance, Figure 4A shows a TDS evaluation of 6 multi-sampled adenomyosis lesions (1-6), adjacent normal endometrium (NE), adjacent normal myometrium (NM), co-occurring leiomyoma (LM) and peripheral blood (B). The highest VAF of a mutation in ARHGAP42 was detected in adenomyosis sample 1 (see Table S17; 3.884% VAF in the sample 8A1), which represents 100% and is shown in darkest blue. Mutations in ARHGAP with relatively high VAF values were detected in adenomyosis samples 3 and 5 (see Table S17; 1.571 and 1.601% in the sample 8A3 and 8A5, respectively), and are shown in a slightly lighter shade of bright blue in these columns. Marginal VAF values for ARHGAP mutations were detected in the remaining adenomyosis samples as well as in non-adenomyosis samples (NE, NM, LM and B), which therefore are shown in white in these columns. In our revised manuscript, in the legend of Figure 3 and 4 on Pages 49 and 51, respectively, we have replaced “Ratio (%)” with “Relative VAF(%)” and state that: “VAF values are shown using a color scale where dark blue is the maximum VAF value within the total sample and white is the absence of a mutation in that gene. The VAF of a mutation in a sample relative to the highest VAF value is represented as a shade of a bright blue color.”

Reviewer-2 comment on rebuttal: I do not understand the rationale for computing and displaying relative VAFs. The purpose of a VAF is to document the variant allele frequency for a specific gene within an individual sample. There is no scientific basis for re-setting the highest VAF for a gene as “100%” and using it as a standard for calculating relative VAFs for that gene in other samples (in

the example provided in the rebuttal above, the highest VAF of a mutation in ARHGAP42 was detected in adenomyosis sample 1 and was only 3.884% but this value was changed to 100% to calculate a “relative” VAF for ARHGAP42 mutations in other samples). The conversion of actual VAFs to relative VAFs serves to artificially inflate the findings and potentially confuse/mislead the readership and the field.

9) Page 11, line 1 states “Some mutations were shared among almost all multi-regional samples from a single patient [e.g : ZNF672 in #28 (Figure 4 3A); C1QTNF and MSS51 in #29 (Figure 3B)], suggesting that these genetic alterations were acquired early during adenomyosis development, while other mutations were more restricted (Figure 3). Most mutations detected in adenomyosis and co-existing leiomyoma were mutually exclusive (Figures 3B–C, 4A, Tables S15–17), implying the lack of a clonal relationship between these disorders. Thus, despite their frequent co-occurrence, adenomyosis and leiomyoma are distinct entities.” Comment: The statistical analyses provided for variants in Tables S15, S16 and S17 (column O) seems to indicate that most mutations are not considered somatic. Therefore, it is unclear if the authors are assigning a different interpretation to variants when they are displayed in Tables S15, S16 and S17 versus when they are displayed in the boxes in Figure 3 and Figure 4. These points require clarification in the manuscript.

We apologize for not presenting our statistical analyses clearly. These analyses apply to not only Tables S15–17 but also Tables S12, 14, 18–20 and 28–29. However, we respectfully disagree with the reviewer most mutations are not considered somatic. In our revised Materials and Methods section, we state that: “As the VAF values of the observed mutations in adenomyosis were quite low, we performed a statistical calculation to determine whether these alterations were true somatic mutations or NGS noise. In general, it is expected that the distribution of the difference in VAFs between lesion and control samples follows a normal distribution. For each position i , a difference of VAFs between adjacent (a) and normal (n) is defined by $d_{ij} = \max_{y \in \Sigma} |a_{y,i} - n_{y,i}|$, where Σ is the set of nucleotides {A,C,G,T}. Let μ_j and σ_j be the mean (shown in column M in Table S12 and column L in Tables S14–20 and S28–29) and the standard deviation (shown in column N in Table S12 and column M in Tables S14–20 and S28–29), respectively, of the set $\{d_{kj} \mid k=j-K, \dots, j-1, j+1, \dots, j+K\}$ for the estimated position j . If one assumes that $X \sim N(\mu_j, \sigma_j^2)$, then the somatic mutation rate for j is defined by $P(X > d_{ij}) < 0.05$, $d_{ij} > 0.001$ and $n_{y,j} < 0.01$, where $y \in \Sigma$. For K values, we used the values shown in length (bp) column O in Table S12 and column N in Tables S14–20 and S28–29 for the individual mutation analyses. For assessment of statistical significance of somatic mutations, the p-value of each mutation is shown in column P in Table S12 and column O in Tables S14–20 and S28–29.” (Page 29, Line 5).

Reviewer-2 comment on rebuttal: This rebuttal requires review by a statistician or computational biologist.

10) The manuscript states “KRAS mutations were more frequent in patients who had been pretreated with the PR agonist DNG17,22 ($p = 0.0002$, Table 1), raising the possibility that DNG is less efficacious in KRAS-mutated disease.” The latter comment is not warranted since the authors do not provide any data on treatment outcome (efficacy) of DNG treatment for their adenomyosis patients. They also do not clarify whether these patients received DNG treatment for their adenomyosis or for co-existing endometriosis. If the treatment was based on co-existent endometriosis this may actually be the feature that is driving the enrichment for KRAS mutations in DNG-treated adenomyosis cases, since this study reports shared KRAS mutations between co-existing adenomyosis and endometriosis.

We thank the reviewer for these thoughtful comments and solid suggestions. We agree that our statement that DNG may be less efficacious in KRAS-mutated disease is based on our in vitro studies of immortalized cells (original Figure 5 and new Figure 6) and has yet to be validated by analyzing the clinical outcomes of the DNG-treated adenomyosis patients in this study. Efficacy of

DNG as a treatment for adenomyosis has been previously reported (references 34 and 35). However, in our study, the clinical assessments of DNG-treated adenomyosis patients were carried out at two independent hospitals with differing assessment protocols and equipment. We felt that combining clinical data from these two different hospitals with differing approaches might be misleading. We totally concur with the reviewer that such correlations properly done are necessary for the translation of our study's findings to clinical application. However, we feel such detailed validation is beyond the scope of our present genomic study. A future validation study that is based on appropriate data collection from a reasonably sized cohort should be performed. In our revised Discussion, we state that: "Our work using an immortalized cell system in vitro has suggested that adenomyotic lesions frequently contain KRAS mutations that may epigenetically downregulate PR and thus reduce DNG efficacy. This observation should be followed up in a large number of DNG-treated adenomyosis patients to determine if our hypothesis is valid and if its implications are translatable to the clinic." (Page 24, Line 1).

Reviewer-2 comment on rebuttal: The rebuttal notes the "statement that DNG may be less efficacious in KRAS-mutated disease is based on our in vitro studies of immortalized cells" and the authors provide a new sentence to clarify this point in the discussion. Therefore, it is unclear why the Results section contains the same information as the original submission, but split into two sentences: "KRAS mutations were more frequent in lesions of patients who had been pretreated with the PR agonist DNG17,25 ($p = 0.0002$) but not with GnRHa (Tables 1 and S22). This observation raises a possibility that DNG may be less efficacious in patients bearing KRAS-mutated adenomyotic clones, allowing these abnormal cells to persist until surgery.." (page 17, lines 11-15).

In answer to the reviewer's second point, DNG was administered to adenomyosis patients without (3 of 22; 13.6%) and with (10 of 48; 20.8%) co-occurring endometriosis. A Fisher exact t-test analysis confirmed that there was no significant difference between these two groups (p -value: 0.7415). Thus, we do not think that DNG treatment drives any enrichment of KRAS mutations in DNG-treated adenomyosis cases.

Reviewer-2 comment on rebuttal: The rebuttal provides the frequency of DNG treatment for patients with and without concurrent endometriosis. However, the question raised in the second point was whether endometriosis may actually be the feature that is driving the enrichment for KRAS mutations in DNG-treated adenomyosis cases, since this study reports shared KRAS mutations between co-existing adenomyosis and endometriosis. Thus, the key issue that still needs to be addressed is whether the frequency of KRAS mutations in adenomyosis differs significantly between DNG-treated patients with and without concurrent endometriosis.

11) Table 1 shows the frequency of DNG treatment for KRAS wildtype versus KRAS mutated cases. The reader has to use these data to manually calculate the frequency of KRAS mutations in patients pretreated with DNG (11 of 13, 84%) and in untreated patients (15 of 57, 26%).

We apologize for not presenting our Table 1 data more conveniently. As 26 different patient characteristics were analyzed in Table 1, we opted to present only DNG-treated (and not untreated) patient numbers in order to summarize the data. In our revised manuscript, we include not only positive (DNG-treated, PR-negative or EN co-occurrence) but also negative (DNG-untreated, PR-positive, or adenomyosis alone) clinical characteristics that showed statistically significant differences. These data now appear in new Table S22.

Reviewer-2 comment on rebuttal: Response is okay, but it might be simpler to just state in the text that the frequency of KRAS mutations was 84% (11 of 13) in patients pretreated with DNG versus 26% (15 of 57) in untreated patients.

12) The concentration of DNG used in the cell-based experiments is not provided. The vehicle (DMSO? Other?) in which DNG was resuspended is not provided. For the cell-based assays the

comparison seems to be DNG-treated cells versus untreated cells. It should be DNG-treated versus vehicle-treated cells. Until the data on DNG- and vehicle-treatments data are provided the validity of the cell-based assays cannot be determined.

We thank the reviewer for this suggestion and apologize for not clarifying that our control was indeed vehicle (DMSO) treatment. In our revised manuscript, we include this information in the legend of new Figure 6 on Page 54 and also note that, for panel B, results were expressed as the percentage of viable cells relative to vehicle control cells. Similarly, for panel C, results were the percentage of BrdU+ cells relative to vehicle control cells.

Reviewer-2 comment on rebuttal: Response is okay

13) The Life Sciences study design information states "Data reproducibility was examined by 3 independent experiments in cell viability (Figure 5A)." Were technical replicates included within each of the three experiments?

We thank the reviewer for this opportunity to improve our data reproducibility. For original Figure 5A, we counted cell numbers in triplicate for each of three independent experiments. However, we recognize that this number of independent experiments is too low to provide convincing reproducibility in the cell line data at the original manuscript. After the submission of our original manuscript, we performed additional independent experiments and now confirm that our data are reproducible. We present these additional data in a new Figure 6 on Page 54 and clarify the number of experiments and replicates in its legend.

Reviewer-2 comment on rebuttal: Response is okay for Figure 6. Are there any other experiments in the manuscript for which the reproducibility is questionable?

14) The data displayed in Figure S16 should be included in the main Figure 5 since a major claim of the paper is that mutant KRAS downregulates PR expression. Having said that, I do not find the observed downregulation of PR expression by KRAS-mut (Figure 16B) very convincing. What exactly are the P values for these data? The fold difference in expression seems to shift from ~1.0 to ~0.5.

We thank the reviewer for this helpful suggestion. In our revised manuscript, we now present the data of original Figures 5 and S16 as a revised Figure 6. We apologize for not presenting our qPCR results clearly. The data are levels of PR-A/B mRNA expression in cells expressing KRAS-WT, KRAS-Mut, PIK3CA-WT or PIK3CA-Mut relative to that in cells transfected with empty vector (control samples). Thus, the mean values and standard deviations represent the fold change in PR-A/B mRNA level compared to that in controls (empty vector transfected samples). The statistical significance (p-values) of differences in these values was assessed by the Welch's t-test. To increase our data reproducibility, we have repeated our qPCR analyses. In our revised manuscript, we present qPCR analyses of triplicate samples in each of six independent experiments. We have confirmed that there is statistically significant downregulation of PR-A/B mRNA in KRAS-Mut (G12C or G12D)-expressing samples compared to empty vector-transfected or KRAS-WT-expressing samples as assessed by the Welch's t-test (p-values < 0.05). The raw data underlying these statistical analyses are included in the Source Data file. In our revised manuscript, we have refined the layout of new Figure 6D and state in its legend: "(D) Quantitative RT-PCR determination of mRNA levels of PR-A/B in the immortalized cells in (A). Values were normalized to GAPDH. Data are the mean fold change + SD relative to levels in control cells stably infected with empty vector. Six independent experiments, each with three technical replicates per group. *p < 0.05 by Welch's t-test. Source data are provided as a Source Data file." (Page 55).

Reviewer-2 comment on rebuttal: Response is okay. However, additional comments are:

1) Since KRAS has oncogenic activity when amplified or mutated in cancer, the authors need to comment on why exogenous expression of a wildtype-KRAS construct (i.e. simulating

amplification) has a different effect on PR levels than exogenous expression of mutant-KRAS constructs. Are the authors certain that the differences observed between wildtype and mutant KRAS are not technical artifacts related to stable transfection issues? Often times, stable transfectants continue to grow in selective medium but they have lost expression of the construct. Were the protein lysates used for the Western in Figure 6A extracted from the same batches of cells as the RNAs used for the qPCR experiment in Figure 6D? In other words, to trust the data in Figure 6D, the authors need to convincingly demonstrate that the batches of cells that were used for the qPCR experiments exogenously expressed equivalent levels of KRAS for WT, G12C, and G12D.

2) Page 19 line 6-13 refers to direct regulation of PR expression by KRAS. However, the experiments provided are not designed to discriminate direct from indirect effects.

3) A limitation that should be acknowledged is that the experiments in Figure 6 were only performed in a single cell line.

15) There are numerous errors and inconsistencies in the descriptions of Table S1-S12. The authors need to carefully check all tables descriptions for accuracy. As examples:

i) Tab for Table S3 is "Table S3 clinical info", whereas the header for Table S3 is "Table S3. Method used to detect genomic alterations in tissue samples from all patients."

ii) Tab for Table S12 is "Table S12 PTENARID1Adeepseq", whereas the header for Table S12 is "Table S12. Summary of TDS read information for pathogenic mutations defined by the ClinVar database."

We apologize for our careless mistakes in listing our Supplementary Tables. We corrected these errors in our revised manuscript.

Reviewer-2 comment on rebuttal: Response is okay.

Response to Reviewer #1

We thank for the reviewer for his/her general comment: “**The authors have significantly enhanced their paper, by adding new data and addressing many of the comments made by both reviewers.**” and “**the results add significantly to the field and are highly novel for adenomyosis**”. Once again, we truly appreciate the constructive and solid experimental suggestions provided during the previous reviewing round, which we agree have markedly improved our manuscript.

Reviewer #1's Specific Comments

Major comments

1) They new analyses, presented in a new figure are a very good addition to the paper. They show an apparent significant increase in KRAS mutational load comparing NE from adenomyosis cases vs. controls, although this was not the case for PIK3CA or PPP2R1A. **This should be made clearer on page 16, lines 10-13. ‘Importantly, KRAS mutations were more often detected in group A and group E than in group Non-A/E (P=xxxx), BUT NOT IN PIK3CA OR PPP2R1A, supporting our hypothesis that an increased frequency of KRAS-mutated clones in NE might be the origin of adenomyosis’.**

Response: We thank the reviewer for this helpful suggestion. As the reviewer appreciated, the frequencies of adenomyosis (group A: 10/18=55.6%) and endometriosis (group E: 7/14=50%) patients with *KRAS*-mutated clones in adjacent histologically normal endometrium are apparently increased compared to patients with neither adenomyosis nor endometriosis. (Non-A/E: 7/24=29.1%). However, this difference was not statistically significant when analyzed using the Fisher exact *t*-test ($p=0.117$ for group A vs. group Non-A/E; $p=0.297$ for group E vs. group Non-A/E), most likely because our cohort size was too small ($n=56$). That being said, our VAF analysis shown in Figure 5B revealed a statistically significant increase in *KRAS*-mutated clones in group A compared to group Non-A/E ($p=0.008$). Although we accept that our

current dataset is not large enough to come to a firm conclusion, we believe that our results support our hypothesis and are worthy of mention. We have noted in our revised manuscript that future work involving a larger, more appropriately sized cohort will be necessary to validate our theory. We state: “**Importantly, we found that *KRAS* mutations, but not *PIK3CA* or *PPP2RIA* mutations, were more often detected in samples from group A (p=0.117) and group E (p=0.297) than in those from group Non-A/E. Although corroboration using a suitably large number of additional patients is required, this observation supports our hypothesis that an increase in the frequency of *KRAS*-mutated clones in NE might be the origin of adenomyosis.**” (Page 16, Line 9).

2) Page 5, line 6: ‘However, no direct evidence to support this hypothesis exists at present due to a lack of genomic analyses of adenomyosis’. It is not just genomic analysis that could elucidate common molecular mechanisms of pathogenesis. This should be rephrased.

Response: We agree with the reviewer’s point. In our revised manuscript, we have rephrased our statement as follows: “**However, no direct evidence to support this hypothesis exists at present due to a lack of knowledge on the molecular mechanisms underlying adenomyosis pathogenesis.**” (Page 5, Line 7).

3) Throughout the manuscript: there is frequent use of the phrase ‘driver mutations’ for adenomyosis or endometriosis. I have a problem with the extensive and definitive use of ‘driver’, as it seems to imply that the mutations are a crucial part of pathogenesis. Yet, they could equally be a consequence/by-product of disease, e.g. as a result of aberrant endometrial proliferation. The author correctly edited this on page 15, line 1 where they state more carefully that the data ‘raise the possibility’ that *KRAS*-mutated clones arising in NE acquire invasiveness and proliferative capacity that enable them to grow ectopically, driving

adenomyosis. Other instances of reference to ‘driver’ mutations should be reworded.

Response: We apologize for our overstatement and agree with the reviewer on this point. We have changed “driver mutation” to context-appropriate descriptions in our revised manuscript (**Page 7, Line 7; Page 11, Line 13; Page 15, Line 9; Page 16, Line 15 and Page 23, Line 16**).

Response to Reviewer #2

We thank for the reviewer for his/her positive comment: **“the authors have responded satisfactorily to most comments, but there are still a few outstanding points, and new points, that need to be addressed.”**. We appreciate the insights and constructive suggestions provided by the reviewer.

Reviewer #2’s Specific Comments

1) Page 7, line 16 states “Our robust criteria and validation by targeted deep sequencing (TDS) (see Materials and Methods, 1 Figure S3) permitted us to detect 134 unique synonymous and non-synonymous single nucleotide variations (SNVs) in 31/51 (60.8%) adenomyosis cases, defining adenomyosis as a clonal disorder with somatic mutations (Tables S5–6).” Reviewer comment: The conclusion that adenomyosis is a “clonal” disorder is not supported by the aforementioned observation.

Reviewer-2 comment on rebuttal: Revision is okay, but please check the call-out to Tables S5-6 in the sentence above; I believe this should be Tables S6-7. Please check the accuracy of all other display item call-outs in the text.

Response: We thank the reviewer for prompting us to check our Table numbering. Although Tables S5–6 do indeed contain the correct information, we recognize that our use of the joint call-out was confusing. Table S5 shows the primer sequences for TDS, whereas Table S6 shows the actual results of our TDS analyses in adenomyosis. To remove any confusion, we now state in our revised manuscript: **“Our robust criteria and validation by TDS (see Materials and Methods, Figure S3, Table S5) permitted us to detect 134 unique**

synonymous and non-synonymous single nucleotide variations (SNVs) in 31/51 (60.8%) adenomyosis cases (Table S6), raising the possibility that adenomyosis is a clonal disorder with somatic mutations.” (Page 9, Line 3).

2) The authors need to indicate what the variable blue coloring symbolizes in the Tables shown in in Figure 3 and 4.

Reviewer-2 comment on rebuttal: I do not understand the rationale for computing and displaying relative VAFs. The purpose of a VAF is to document the variant allele frequency for a specific gene within an individual sample. There is no scientific basis for re-setting the highest VAF for a gene as “100%” and using it as a standard for calculating relative VAFs for that gene in other samples (in the example provided in the rebuttal above, the highest VAF of a mutation in ARHGAP42 was detected in adenomyosis sample 1 and was only 3.884% but this value was changed to 100% to calculate a “relative” VAF for ARHGAP42 mutations in other samples). The conversion of actual VAFs to relative VAFs serves to artificially inflate the findings and potentially confuse/mislead the readership and the field.

Response: We apologize for not presenting our multi-regional mutation analyses properly. In response to the reviewer’s criticisms, we now show actual percentage VAF values in revised **Figures 3, 4, S13F, S15B and S15D**. We have relabelled the color scale.

3) The manuscript states “KRAS mutations were more frequent in patients who had been pretreated with the PR agonist DNG17,22 ($p = 0.0002$, Table 1), raising the possibility that DNG is less efficacious in KRAS-mutated disease.” **The latter comment is not warranted since the authors do not provide any data on treatment outcome (efficacy) of DNG treatment for their adenomyosis patients.** They also do not clarify whether these patients received DNG treatment for their adenomyosis or for co-existing endometriosis. If the treatment was based on co-existent endometriosis this may actually be the feature that is driving the enrichment for KRAS mutations in DNG-treated adenomyosis cases, since this study reports shared KRAS

mutations between co-existing adenomyosis and endometriosis.

Reviewer-2 comment on rebuttal: The rebuttal provides the frequency of DNG treatment for patients with and without concurrent endometriosis. However, the question raised in the second point was whether endometriosis may actually be the feature that is driving the enrichment for *KRAS* mutations in DNG-treated adenomyosis cases, since this study reports shared *KRAS* mutations between co-existing adenomyosis and endometriosis. Thus, the key issue that still needs to be addressed is whether the frequency of *KRAS* mutations in adenomyosis differs significantly between DNG-treated patients with and without concurrent endometriosis.

Response: We apologize for not addressing the reviewer's original question properly. In response to the reviewer's suggestion, we investigated the statistical significance of differences in the frequency of *KRAS* mutations between DNG-treated patients with or without co-occurring endometriosis. Although, as expected, there was no statistically significant difference between these two groups ($p=0.4231$; 2/3 patients without endometriosis vs. 9/10 patients with endometriosis), we feel that our cohort size ($n=13$) was too small and that there is a need to assess this question using a larger patient population in the future. Thus, we accept this criticism and state in our revised manuscript: **“This observation generates two hypotheses: 1) DNG may be less efficacious in patients bearing *KRAS*-mutated adenomyotic clones, allowing these abnormal cells to persist until surgery; and 2) DNG treatment drives an enrichment of *KRAS* mutations in adenomyosis.”** (Page 17, Line 16); and **“Thus, expression of mutated *KRAS* interferes with DNG's ability to reduce cell proliferation. Taken together, these data support our hypothesis #1 (*KRAS*-mutated adenomyotic clones are less sensitive to DNG treatment) rather than hypothesis #2 (DNG drives *KRAS* mutations in adenomyosis individuals). However, it should be noted that future follow-up studies using an appropriately sized cohort will be necessary to reach a definitive conclusion.”** (Page 18, Line 16).

4) Table 1 shows the frequency of DNG treatment for KRAS wildtype versus KRAS mutated cases. The reader has to use these data to manually calculate the frequency of KRAS mutations in patients pretreated with DNG (11 of 13, 84%) and in untreated patients (15 of 57, 26%).

Reviewer-2 comment on rebuttal: Response is okay, but it might be simpler to just state in the text that the frequency of KRAS mutations was 84% (11 of 13) in patients pretreated with DNG versus 26% (15 of 57) in untreated patients.

Response: We thank the reviewer for this helpful suggestion. In our revised manuscript, we state: **“Specifically, KRAS mutations were more frequent in lesions of patients who had been pretreated with DNG [84% (11/13)] compared to non-pretreated patients [26% (15/57); $p = 0.0002$], a difference not found in patients treated with GnRHa (Tables 1 and S22).” (Page 17, Line 14).**

5) Since KRAS has oncogenic activity when amplified or mutated in cancer, the authors need to comment on why exogenous expression of a wildtype-KRAS construct (i.e. simulating amplification) has a different effect on PR levels than exogenous expression of mutant-KRAS constructs. Are the authors certain that the differences observed between wildtype and mutant KRAS are not technical artifacts related to stable transfection issues? Often times, stable transfectants continue to grow in selective medium but they have lost expression of the construct. Were the protein lysates used for the Western in Figure 6A extracted from the same batches of cells as the RNAs used for the qPCR experiment in Figure 6D? In other words, to trust the data in Figure 6D, the authors need to convincingly demonstrate that the batches of cells that were used for the qPCR experiments exogenously expressed equivalent levels of KRAS for WT, G12C, and G12D.

Response: We thank the reviewer for these thoughtful comments. Indeed, the same batches of cells were used to produce the data in Figures 6A and 6D. We apologize for not clearly presenting our qPCR data on the levels of exogenous expression of our KRAS constructs. As can be seen by perusing our raw qPCR data (**shown in the Source Data file, related to Figure 6D**), we observed ~3-fold overexpression (compared to

empty vector-expressing controls) of not only *KRAS*-WT, G12C and G12D but also *PIK3CA*. As the reviewer points out, exogenous *KRAS*-WT expression can mimic *KRAS* amplification, but under our experimental conditions, only overexpression of exogenous *KRAS*-Mut (and not *KRAS*-WT) was able to significantly downregulate *PR* mRNA.

6) Page 19 line 6-13 refers to direct regulation of PR expression by KRAS. However, the experiments provided are not designed to discriminate direct from indirect effects.

Response: We accept the reviewer's insightful criticism on this point. Our current data discriminate between cause vs. consequence rather than between direct vs. indirect effects. In our revised manuscript, we now state: **“To investigate whether the reduced PR protein in *KRAS*-mutated adenomyosis samples was due to an effect of mutant *KRAS* on *PR* mRNA expression, we applied qPCR analysis to our immortalized endometrial cells overexpressing *KRAS*-Mut, *KRAS*-WT, *PIK3CA*-WT or *PIK3CA*-Mut (please see the Source Data file). We found that *PR*-A/B mRNAs were indeed downregulated in cells expressing *KRAS*-Mut but not in those expressing *KRAS*-WT, *PIK3CA*-WT or *PIK3CA*-Mut (Figure 6D). Although these *in vitro* overexpression data derived from a single cell line do not show whether *KRAS* mutation downregulates *PR* mRNA directly or indirectly, they suggest that *KRAS*-Mut is linked to downregulation of *PR* expression. Further experiments employing additional independent cell lines and/or patient-derived primary cultures should be conducted to validate our results.”** (Page 19, Line 15).

7) A limitation that should be acknowledged is that the experiments in Figure 6 were only performed in a single cell line.

Response: We thank the reviewer for this helpful suggestion. As noted above, our revised manuscript now states: **“Although these *in vitro* overexpression data derived from a single cell line do not show whether**

***KRAS* mutation downregulates *PR* mRNA directly or indirectly, they suggest that *KRAS*-Mut is linked to downregulation of *PR* expression. Future work employing additional independent cell lines and/or patient-derived primary cultures should be conducted to validate our results.” (Page 20; Line 5).**

REVIEWERS' COMMENTS:

Reviewer #1 (Remarks to the Author):

The authors described the results as follows in response to my query regarding non-significance of the KRAS results: 'Importantly, we found that KRAS mutations, but not PIK3CA or PPP2R1A mutations, were more often detected in samples from group A ($p=0.117$) and group E ($p=0.297$) than in those from group Non-A/E. Although corroboration using a suitably large number of additional patients is required, this observation supports our hypothesis that an increase in the frequency of KRAS-mutated clones in NE might be the origin of adenomyosis.' (Page 16, Line 9).'

It is very important that results that are non-significant are not described as if they show a difference, even though they 'look' as if they are different. Please revise as follows: 'We found that KRAS mutations were not significantly more frequent in samples from group A ($p=0.117$) and group E ($p=0.297$) than in those from group Non-A/E. However, our VAF analysis revealed a statistically significant increase in KRAS-mutated clones in group A compared to group Non-A/E ($p=0.008$). Follow-up in a suitably large number of additional patient samples is required, but if corroborated, the latter result suggests our hypothesis that an increase in the frequency of KRAS-mutated clones in NE might be the origin of adenomyosis.' (Page 16, Line 9).'

Reviewer #2 (Remarks to the Author):

The Authors have provided satisfactory responses to the previous round of comments.

Response to Reviewer #1

We truly appreciate the constructive suggestions provided during the previous reviewing rounds.

Reviewer #1's Specific Comments

The authors described the results as follows in response to my query regarding non-significance of the KRAS results: ‘Importantly, we found that KRAS mutations, but not PIK3CA or PPP2R1A mutations, were more often detected in samples from group A ($p=0.117$) and group E ($p=0.297$) than in those from group Non-A/E. Although corroboration using a suitably large number of additional patients is required, this observation supports our hypothesis that an increase in the frequency of KRAS-mutated clones in NE might be the origin of adenomyosis.’ (Page 16, Line 9).’

It is very important that results that are non-significant are not described as if they show a difference, even though they ‘look’ as if they are different. Please revise as follows:

‘We found that KRAS mutations were not significantly more frequent in samples from group A ($p=0.117$) and group E ($p=0.297$) than in those from group Non-A/E. However, our VAF analysis revealed a statistically significant increase in KRAS-mutated clones in group A compared to group Non-A/E ($p=0.008$). Follow-up in a suitably large number of additional patient samples is required, but if corroborated, the latter result suggests our hypothesis that an increase in the frequency of KRAS-mutated clones in NE might be the origin of adenomyosis.’ (Page 16, Line 9).’

Response: We thank the reviewer for this helpful suggestion. We agree with the reviewer’s point. In our revised manuscript, we have revised our statement as the reviewer recommended (**Page 16, Line 12– Page 17, Line 2**).

Response to Reviewer #2

Response: We thank for the reviewer for his/her positive comment: “**The Authors have provided satisfactory responses to the previous round of comments**”. We are thankful for constructive feedback provided by the reviewer during the review process.